# LC3-associated phagocytosis of neutrophils triggers tumor ferroptotic cell death in glioblastoma

Tong Lu[1], Patricia P Yee[1,2,3], Stephen Y Chih [ID][1,2], Miaolu Tang [ID][1], Han Chen[4], Dawit G Aregawi[5,6,7], Michael J Glantz[5,6,8], Brad E Zacharia[5,6,9], Hong-Gang Wang[1,6,10] & Wei Li [ID][1,6,11 ✉]

## Abstract

**Necrosis in solid tumors is commonly associated with poor prognostic but how these lesions expand remains unclear. Studies have found that neutrophils associate with and contribute to necrosis development in glioblastoma by inducing tumor cell ferroptosis through transferring myeloperoxidase-containing granules. However, the mechanism of neutrophilic granule transfer remains elusive. We performed an unbiased small molecule screen and found that statins inhibit neutrophil-induced tumor cell death by blocking the neutrophilic granule transfer. Further, we identified a novel process wherein neutrophils are engulfed by tumor cells before releasing myeloperoxidase-containing contents into tumor cells. This neutrophil engulfment is initiated by integrin-mediated adhesion, and further mediated by LC3-associated phagocytosis (LAP), which can be blocked by inhibiting the Vps34-UVRAG-RUBCN-containing PI3K complex. Myeloperoxidase inhibition or Vps34 depletion resulted in reduced necrosis formation and prolonged mouse survival in an orthotopic glioblastoma mouse model. Thus, our study unveils a critical role for LAP-mediated neutrophil internalization in facilitating the transfer of neutrophilic granules, which in turn triggers tumor cell death and necrosis expansion. Targeting this process holds promise for improving glioblastoma prognosis.**

**Keywords** Tumor Necrosis; LC3-associated Phagocytosis; Neutrophil; Ferroptosis; Glioblastoma
**Subject Categories** Autophagy & Cell Death; Cancer; Immunology

## Introduction

Tumor necrosis describes the extensive cell and tissue death frequently observed in advanced solid tumors. Serving as a histological hallmark and a predictor of poor prognosis in various cancers, necrosis has been implicated in driving tumor progression and resistance to several therapies (Yee and Li, 2021). Therefore, the prevention of necrosis in tumors could yield significant clinical benefits. Despite its significance, the precise mechanisms underlying necrosis in tumor development remain largely unclear, although it is generally believed that necrosis is initiated by specific metabolic stresses resulting from the deprivation of oxygen and nutrients due to inadequate blood supply.

Necrosis in glioblastomas (GBM), the most common and aggressive primary brain malignancy in adults, is a well-known diagnostic hallmark. Previous studies have identified a temporal and spatial correlation between necrosis and neutrophils (Yee et al, 2020). The association between neutrophils and necrosis has also been observed during the GBM proneural to mesenchymal transition and in melanoma metastases (Chen et al, 2023; Weide et al, 2023). This association is consistent with findings that neutrophils undergo pro-tumorigenic reprogramming during GBM progression (Friedmann-Morvinski and Hambardzumyan, 2023; Magod et al, 2021). Neutrophils are not merely passively recruited to necrotic sites within tumors; they also actively contribute to promoting necrosis by inducing tumor cell death through ferroptosis (Yee et al, 2020). In this process, neutrophil-derived myeloperoxidase (MPO) is transferred to tumor cells and responsible for inducing tumor cell ferroptosis. However, the mechanism underlying the transfer of MPO-containing contents remains elusive.

In this study, utilizing an unbiased small molecule screen, we discovered that the statin family of β-Hydroxy β-methylglutaryl-CoA (HMG-CoA) reductase inhibitors can rescue tumor cells from neutrophil-induced cell death. This rescue mechanism involves blocking the internalization of neutrophils by tumor cells, thereby preventing the transfer of neutrophilic contents into tumor cells. Further investigation into the process of neutrophil internalization revealed its occurrence through LC3-associated phagocytosis. Pharmacological or genetic inhibition of the Vps34-containing PI3K complex in tumor cells can suppress their ability to engulf neutrophils, thus protecting themselves from neutrophil-induced

[1]Division of Hematology and Oncology, Department of Pediatrics, Penn State College of Medicine, Hershey, PA, USA. [2]Medical Scientist Training Program, Penn State College of Medicine, Hershey, PA, USA. [3]Department of Neurosurgery, Johns Hopkins Hospital, Baltimore, MD, USA. [4]Transmission Electron Microscopy (TEM) Core, Penn State College of Medicine, Hershey, PA, USA. [5]Division of Neurooncology and Skull Base Surgery, Department of Neurosurgery, Penn State College of Medicine, Hershey, PA, USA. [6]Penn State Cancer Institute, Penn State College of Medicine, Hershey, PA, USA. [7]Department of Neurology, Penn State College of Medicine, Hershey, PA, USA. [8]Department of Medicine, Penn State College of Medicine, Hershey, PA, USA. [9]Department of Otolaryngology-Head and Neck Surgery, Penn State College of Medicine, Hershey, PA, USA. [10]Department of Pharmacology, Penn State College of Medicine, Hershey, PA, USA. [11]Department of Biochemistry and Molecular Biology, Penn State College of Medicine, Hershey, PA, USA.
✉E-mail: weili@pennstatehealth.psu.edu

cell death. In an orthotopic xenograft GBM mouse model, inhibition of MPO or depletion of Vps34 reduced necrosis formation, leading to prolonged survival of tumor-bearing mice.

# Results

## Statins can inhibit neutrophil-mediated cell-killing

To study neutrophil-induced tumor cell death, we performed a small-molecule screen using a library containing 1190 compounds, most of which are FDA-approved drugs. The screen used a coculture system containing differentiated HL-60 (dHL-60) neutrophils and LN229$^{TAZ(4SA)}$ human GBM cells. As a control, tumor cells cultured alone were also exposed to the compounds (Fig. 1A). The results showed that five hits significantly (FDR < 0.01) increased survival of LN229$^{TAZ(4SA)}$ cells in the coculture. Within these hits, both fluvastatin and atorvastatin belong to statins (Fig. 1B, red circles). Interestingly, among another four statins included in the screen, three were also able to increase the viability of LN229$^{TAZ(4SA)}$ cells in the coculture (Fig. 1B, blue circles). Multiple statins (five out of six) showed positive effects from the screen, suggesting that statins can inhibit the neutrophil-induced tumor cell death.

To confirm the above finding, we tested simvastatin and fluvastatin individually. Indeed, both statins increased the viability of LN229$^{TAZ(4SA)}$ cells when they are cocultured with dHL-60 neutrophils (Fig. 1C). Statins are β-Hydroxy β-methylglutaryl-CoA (HMG-CoA) reductase inhibitors, which can block mevalonate production and multiple downstream effectors of the mevalonate pathway (Mullen et al, 2016). To understand how inhibition of the mevalonate pathway by statins can suppress neutrophil-induced tumor cell death, we tested three downstream branches of the pathway, including squalene-cholesterol production, protein farnesylation and geranylgeranylation, by using inhibitors working on each branch (Fig. 1D). GGTI-298, but not YM-53601 or FTI-277, rescued the tumor cell-killing by neutrophils (Figs. 1E, EV1A and EV1B). Primary neutrophils isolated from tumors can induce tumor cell death through ferroptosis (Yee et al, 2020). To examine whether tumor cell death induced by these tumor-associated neutrophils (TAN) can also be inhibited by statins, we purified TAN from LN229$^{TAZ(4SA)}$ cell-derived tumors and cocultured them with LN229$^{TAZ(4SA)}$ cells. Indeed, the viability of LN229$^{TAZ(4SA)}$ cells was increased by simvastatin, fluvastatin, and GGTI-298, but not YM-53601 (Fig. 1F, G).

It was previously shown that mevalonate pathway inhibition can regulate ferroptosis (Garcia-Bermudez et al, 2019; Liang et al, 2022; Shimada et al, 2016; Viswanathan et al, 2017), and neutrophil-induced cell-killing occurs through ferroptosis (Yee et al, 2020). We observed that dHL-60 cells-induced LN229$^{TAZ(4SA)}$ cell death can be rescued by ferrostatin-1, liproxstatin-1 and deferoxamine (DFO) (Fig. EV1C), therefore, confirming the cell death is ferroptosis (Yee et al, 2020). We then examined whether the mevalonate pathway inhibitors could inhibit ferroptosis. Simvastatin, fluvastatin, YM-53601 and GGTI-298 all inhibited IKE-induced ferroptosis, although FTI-277 did not (Fig. 1H). The ferroptosis inhibitory effect of these small molecules, except FTI-277, was also observed when ferroptosis was induced by RSL3, although simvastatin and fluvastatin showed milder effect (Fig. 1I). Because YM-53601 inhibits both IKE- and RSL3-induced ferroptosis but cannot rescue

neutrophil-induced tumor cell-killing, it is likely that inhibiting ferroptosis per se is not sufficient for rescuing neutrophil-induced cell-killing by simvastatin, fluvastatin, or GGTI-298.

Our previous studies indicated that neutrophils trigger tumor cell ferroptotic death through intercellular transfer of MPO-containing granular contents into tumor cells and that MPO activity is essential for the neutrophil cytotoxicity (Yee et al, 2020). To further examine whether MPO activity is required for neutrophil-mediated cell killing, we used 4-aminobenzoic acid hydrazide (4-ABAH) and PF06282999, two small-molecule MPO inhibitors (Kettle et al, 1995; Ruggeri et al, 2015). Adding either of them can rescue LN229$^{TAZ(4SA)}$ cells from dHL-60-induced cell killing (Fig. EV1D), therefore confirming the requirement of MPO activity in the cell-killing process. We further examined whether simvastatin, fluvastatin, or GGTI-298 can inhibit the neutrophilic content transfer. As reported previously (Yee et al, 2020), when dHL-60 neutrophils prelabelled with a lipophilic membrane fluorescent dye, PKH26, were cocultured with LN229$^{TAZ(4SA)}$ tumor cells, we observed that tumor cells contain PKH26$^+$ puncta and MPO$^+$ granules (Fig. 1J). In the presence of simvastatin, fluvastatin, or GGTI-298, but not YM-53601, tumor cells containing neutrophil-derived PKH26$^+$ puncta and MPO$^+$ granules were much fewer than in controls (Fig. 1J–L). Therefore, these results suggested that several small molecules that interrupt the mevalonate-protein geranylgeranylation pathway can inhibit neutrophil-induced cell-killing, likely through blocking the neutrophilic content transfer.

## Direct cell–cell contact is required for the neutrophilic content transfer and tumor cell-killing by neutrophils

To study how the neutrophilic content transfer is achieved, we examined whether direct cell–cell contact is required for the transfer process. Two approaches were employed for this test (Fig. 2A). First, we compared LN229$^{TAZ(4SA)}$ cells cultured in dHL-60 neutrophil-conditioned medium to those cocultured with neutrophils directly. Although neutrophil-derived PHK26$^{high}$ contents were observed in the cocultured LN229$^{TAZ(4SA)}$ cells, those cultured in the conditioned media did not show these contents (Fig. 2B). In the second approach, we used a Boyden chamber assay to coculture neutrophils and tumor cells. In this case, intercellular transfer of neutrophil-derived PHK26$^{high}$ contents was not observed (Fig. 2B). These observations were also confirmed when LN229$^{TAZ(4SA)}$ cells were cocultured with differentiated 32Dcl3 (d32Dcl3) mouse neutrophils when the same assays were conducted (Fig. 2C). In addition, when MPO was examined through immunofluorescent staining, its transfer to tumor cells from d32Dcl3 neutrophils did not occur when these cells were not cultured together or were cocultured with the Boyden chamber (Fig. 2D). Intracellular lipid peroxide level increases in tumor cells when they are cocultured with neutrophils and this effect can be blocked by ferroptosis inhibitors (Yee et al, 2020). To examine if accumulation of lipid peroxides and cell death in tumor cells also rely on a direct tumor cell-neutrophil contact, we applied two similar approaches as described above. We found that both lipid peroxide accumulation and cell death of LN229$^{TAZ(4SA)}$ cells were markedly diminished in dHL-60- or d32Dcl3-conditioned media, or in the presence of the insert (Fig. 2E–H). Therefore, these results indicated that direct cell contact is required for the neutrophilic content transfer and tumor cell-killing by neutrophils.

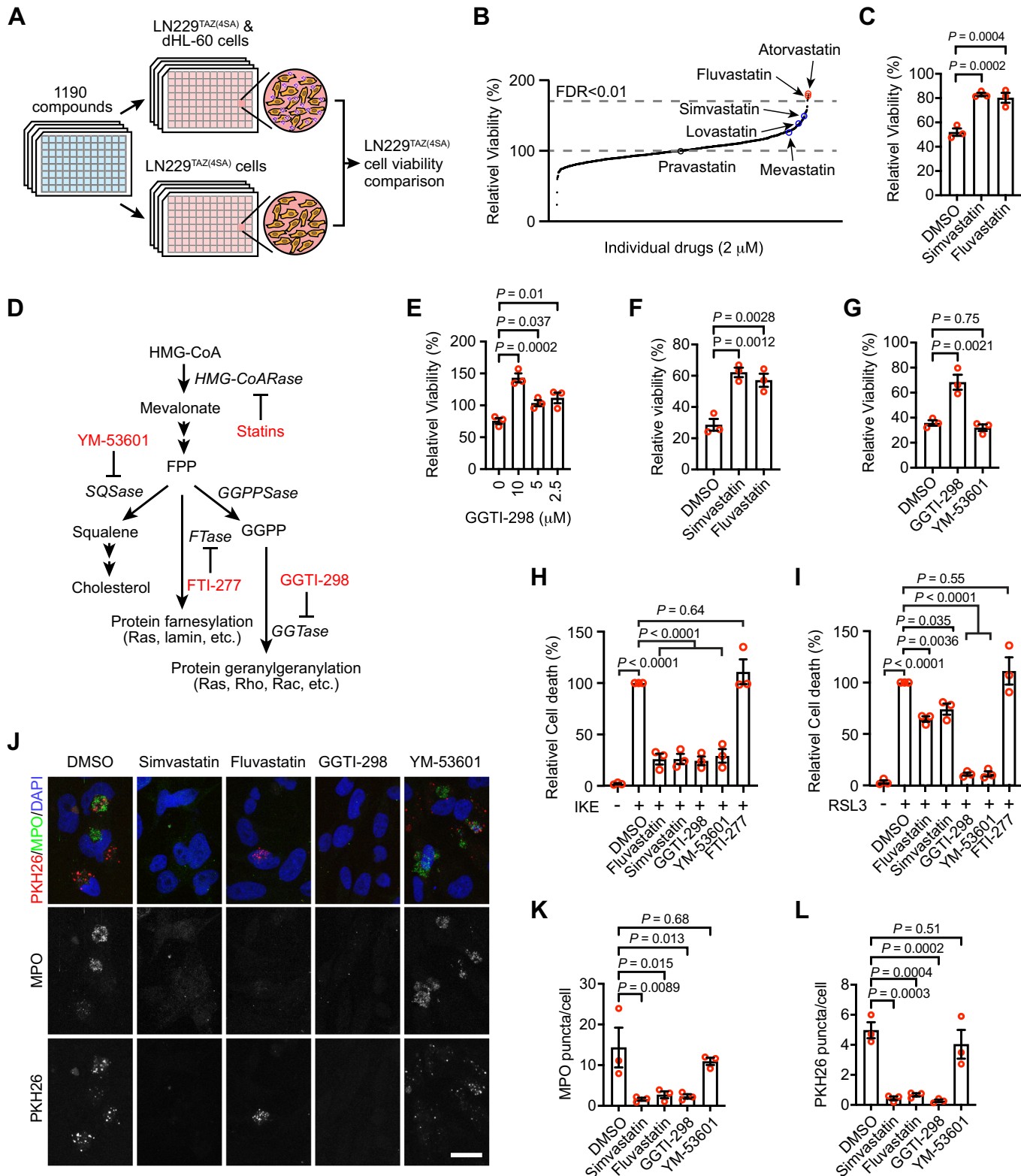

◀ **Figure 1.  An unbiased screen found that statins can inhibit neutrophil-mediated cell killing.**

(A) Schematic detailing the unbiased drug screen. Luciferase readouts of cocultured wells were normalized to their respective monocultured wells, both in the presence of the tested drug (2 µM), to determine the relative viability. (B) Dataset summarizing the results of the unbiased drug screen. Relative viabilities of each drug were normalized to the DMSO control groups. (C) Cancer cell viability determined using luciferase readouts for LN229$^{TAZ(4SA)}$ cells cocultured with dHL-60 cells with DMSO, simvastatin (2.5 µM), or fluvastatin (2.5 µM). Luminescence readouts of cocultured wells were normalized to their respective LN229$^{TAZ(4SA)}$ cells monocultured wells, both with drugs ($n = 3$ independent experiments), one-way ANOVA. (D) Diagram summarizing the MVA pathway and the targets of drugs tested. (E) Luciferase assay results for LN229$^{TAZ(4SA)}$ cells cocultured with dHL-60 cells with GGTI-298 ($n = 3$ independent experiments). Luminescence readouts of cocultured wells were normalized to their respective LN229$^{TAZ(4SA)}$ cells monocultured wells, both with drugs. One-way ANOVA. (F) Cancer cell viability determined using luciferase readouts for LN229$^{TAZ(4SA)}$ cells cocultured with TAN with DMSO, simvastatin (2.5 µM), or fluvastatin (2.5 µM). Luminescence readouts of cocultured wells were normalized to their respective LN229$^{TAZ(4SA)}$ cells monocultured wells, both with drugs ($n = 3$ independent experiments), one-way ANOVA. (G) Luciferase assay results for LN229$^{TAZ(4SA)}$ cells cocultured with TAN with GGTI-298 (10 µM) and YM-53691 (10 µM) ($n = 3$ independent experiments). Luminescence readouts of cocultured wells were normalized to their respective LN229$^{TAZ(4SA)}$ cells monocultured wells, both with drugs. One-way ANOVA. (H, I) Cell death evaluated using YOYO-3 Iodide stain for LN229$^{TAZ(4SA)}$ cells cultured with (H) IKE (0.75 µM) or (I) RSL3 (0.125 µM) with DMSO, fluvastatin (2.5 µM), simvastatin (2.5 µM), GGTI-298 (5 µM), YM-53601 (10 µM), or FTI-277 (10 µM) ($n = 3$ independent experiments), one-way ANOVA. (J) Representative images showing immunofluorescent PKH26, MPO and DAPI signal for LN229$^{TAZ(4SA)}$ cells cocultured with PKH26-labeled dHL-60 cells with DMSO, simvastatin (5 µM), fluvastatin (5 µM), GGTI-298 (10 µM), or YM-53601 (10 µM) ($n = 3$). Scale bar, 10 µm. (K) Quantification of MPO puncta in each tumor cell in the experiments as shown in (J). Two fields of view were used for each group in each experiment ($n = 3$ independent experiments). One-way ANOVA. (L) Quantification of PKH26 puncta in each tumor cell in the experiments as shown in (J). Two fields of view used for each group in each experiment ($n = 3$ independent experiments). One-way ANOVA. Error bars, s.e.m. Source data are available online for this figure.

## Integrin-mediated cell adhesion is required for tumor cell-killing by neutrophils

HL-60 cells in a non-differentiated or naive state (ndHL-60) are not able to induce tumor cell death or transfer their contents into tumor cells (Yee et al, 2020). When PKH26-labeled ndHL-60 cells were cocultured with LN229$^{TAZ(4SA)}$ tumor cells, we found that these killing-incapable cells showed less attachment to tumor cells than their killing-capable counterparts (Fig. 3A,B). The attachment appears to involve certain cell adhesion, as it is resistant to a physical disturbance (i.e., shaking the dishes or rinsing cells) (Fig. 3C,D). To understand the differential attachment capability of HL-60 cells in the two different states, we performed RNA-seq analysis of gene expression in HL-60 cells. The gene expression profile changed remarkably when dHL-60 cells were compared with ndHL-60 cells (Fig. EV2A). Through Ingenuity Pathway Analysis of the differentially expressed genes (FC ≥ 1, FDR < 0.05), we found that the neutrophil degranulation pathway was the most significantly upregulated signaling (Fig. 3E). Genes involved in this pathway showed markedly different expression in the two types of HL-60 cells (Fig. EV2B). Among these genes, integrin family genes (Fig. EV2A), such as *ICAM1*, *ITGAM*, and *ITGB2* are important for neutrophil adhesion to endothelial cells during extravasation, while *ITGAV* could activate other integrin signaling in neutrophils. Integrin $\alpha_v\beta_1$ binds to its ligands through recognition of the RGD motif (Plow et al, 2000).

To examine whether RGD-recognizing integrins are involved in the interaction between neutrophils and tumor cells, we used an RGDS peptide to block the integrin-ligand binding. Consistent with previous studies (Pierschbacher and Ruoslahti, 1984; Plow et al, 2000), adding RGDS into the tumor cell monoculture led to detachment of tumor cells, indicating that tumor cells rely on the RGD-interacting integrin to attach to the culture surface. To circumvent this effect on tumor cell-matrix attachment, we conducted the coculture assay in a 3-dimensional suspension condition. Like the 2-dimensional condition, PKH26-labeled dHL-60 cells adhered to the cocultured tumor cells (Fig. 3F, left panel). Most of the tumor cells appeared to overlap with dHL-60 cells (Fig. 3F,G). In the presence of the RGDS peptide, overlap of the two cell populations was significantly reduced (Fig. 3F,G). Using flow

cytometry analysis in which dHL-60 was pre-labeled by PKH26 and tumor cells expressed ZsGreen, we found that cells possessing both markers were markedly reduced when adding the RGDS peptide (Fig. 3H,I). In this condition, tumor cells showed increased survival compared to the coculture condition without the RGDS peptide (Fig. 3J). When LN229$^{TAZ(4SA)}$ cells were cocultured with TAN, their viability also increased in the presence of the RGDS peptide (Fig. 3K). Therefore, the above results suggested that integrin-mediated adhesion between tumor cells and neutrophils is involved in the cell–cell interaction and neutrophil-induced tumor cell-killing. Since ITGAV is an RGD motif recognition integrin, that is upregulated in dHL-60 cells (Fig. EV2A), we examined whether it is involved in the neutrophil-mediated cell-killing process. Silencing the expression of ITGAV through two shRNAs in dHL-60 cells increased LN229$^{TAZ(4SA)}$ cell viability in the coculture (Figs. EV2C and Fig. 3L). To examine whether any RGD motif recognition integrins in LN229$^{TAZ(4SA)}$ cells could be involved, we knocked down ITGB1 or ITGA5 individually in the tumor cells through shRNAs targeting each of them (Fig. EV2D,E). Depletion each of them did not change the viability of LN229$^{TAZ(4SA)}$ cells cocultured with dHL-60 cells (Fig. EV2F,G). Overall, these results indicated that integrin containing the ITGAV subunit in dHL-60 cells is responsible for the cell–cell interaction and neutrophil-induced tumor cell-killing.

## Neutrophil internalization by tumor cells

To further characterize the process from neutrophil-tumor cell adhesion to neutrophilic content transfer, we carried out time-lapse live-cell imaging of the coculture containing d32Dcl3 neutrophils and LN229$^{TAZ(4SA)}$ tumor cells. The results showed that neutrophils are internalized by tumor cells, then fragmented inside of the latter cells (Fig. 4A, arrows). This phenomenon was also observed when d32Dcl3 neutrophils were cocultured with patient-derived primary GBM cells (Fig. 4B, arrows). With live-cell imaging, we observed that d32Dcl3 neutrophil internalization into LN229$^{TAZ(4SA)}$ tumor cells was followed by tumor cell death (Fig. 4C, arrowheads). We then used transmission electron microscopy (TEM) to further examine the internalization process. The results showed that d32Dcl3 neutrophils use their membrane protrusion to establish cell adhesion with LN229$^{TAZ(4SA)}$ tumor cells through alignment of

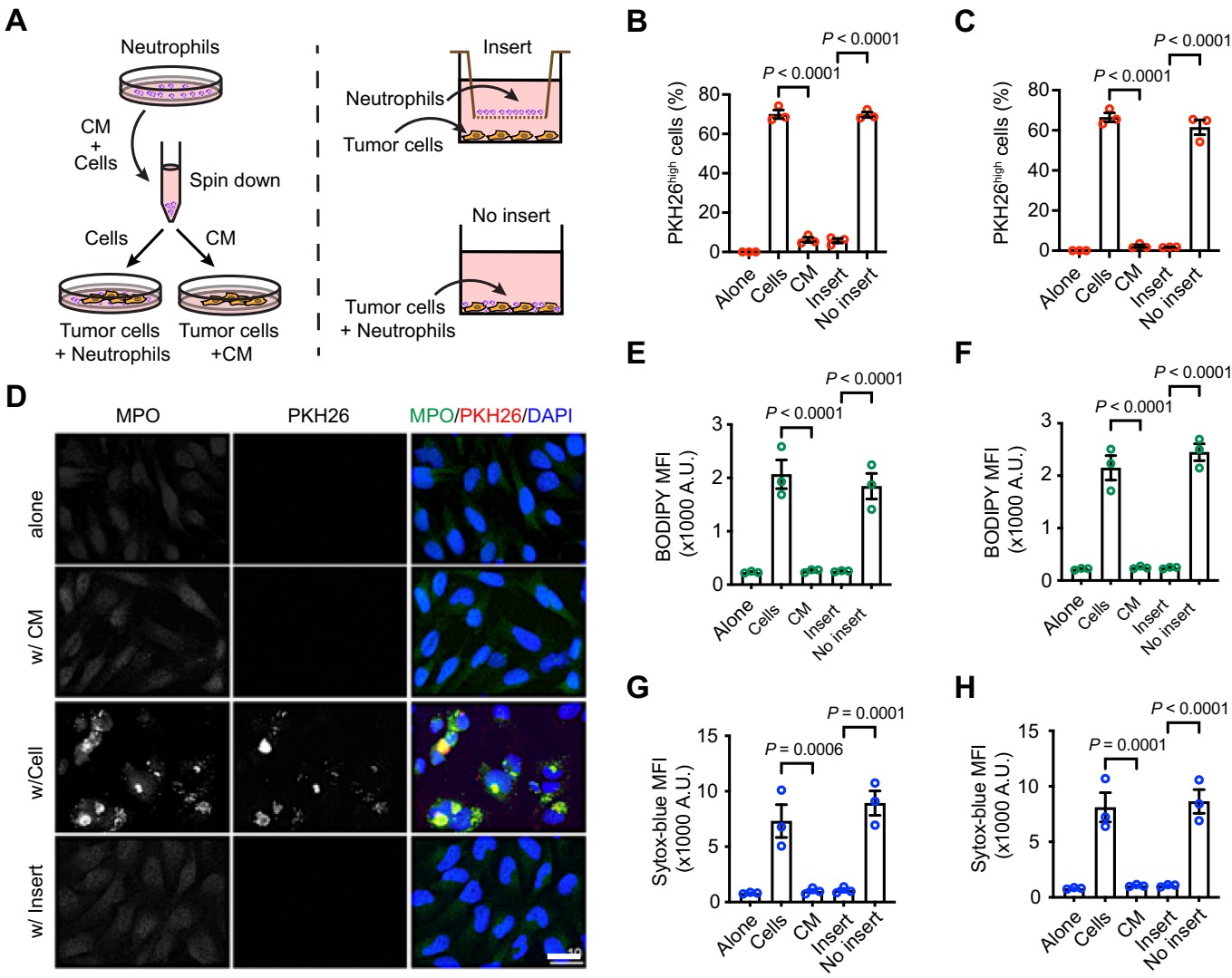

**Figure 2.  Direct cell–cell contact is required for the neutrophilic content transfer and tumor cell-killing by neutrophils.**

(**A**) Diagram showing the two strategies of coculturing neutrophils and tumor cells. On the left, ZsGreen-expressing LN229$^{TAZ(4SA)}$ cells were cocultured with PKH26-labeled neutrophils (dHL-60 or d32Dcl3cl3) (cells) or with PKH26-labeled neutrophil-conditioned media (CM). On the right, ZsGreen-expressing LN229$^{TAZ(4SA)}$ cells were cocultured with PKH26-labeled neutrophils separated by a Boyden chamber insert (3 µm) or with PKH26-labeled neutrophils without the insert (no insert). (**B, C**) Flow cytometry analysis of PKH$^{High}$ ZsGreen$^+$ cells in the coculture as shown in (**A**). One-way ANOVA. Neutrophils used were dHL-60 cells (**B**) and d32Dcl3cl3 cells (**C**) ($n = 3$ independent experiments for each). (**D**) Representative images showing immunofluorescent PKH26 signal, MPO staining, and DAPI staining. Groups shown are similar to (**C**) ($n = 3$), except that LN229$^{TAZ(4SA)}$ cells do not express ZsGreen. Scale bar, 10 µm. (**E, F**) Flow cytometry analysis of groups indicated in (**B**) and (**C**), respectively, for BODIPY. LN229$^{TAZ(4SA)}$ cells do not express ZsGreen in this case ($n = 3$ independent experiments for each). One-way ANOVA. (**G, H**) Flow cytometry analysis of groups indicated in (**B**) and (**C**), respectively, for Sytox-blue ($n = 3$ independent experiments for each). One-way ANOVA. Error bars, s.e.m. Source data are available online for this figure.

their membranes (Fig. 4D, arrowheads). Similar cell membrane alignment was also observed between dHL-60 neutrophils and LN229$^{TAZ(4SA)}$ tumor cells (Fig. EV3A, arrowheads). TEM also confirmed the existence of complete d32Dcl3 or dHL-60 neutrophils inside of LN229$^{TAZ(4SA)}$ tumor cells when they are cocultured (Fig. 4E, left panel, and Fig. EV3B, arrows). In brain tissue sections derived from mice bearing LN229$^{TAZ(4SA)}$ tumors, we also observed similar neutrophil internalization by tumor cells (Fig. 4E, right panel, arrow). Electron microscopy of LN229$^{TAZ(4SA)}$ tumor cells in vivo or cocultured with dHL-60 neutrophils in vitro revealed that tumor cells contain two types of granules. One type has a larger

diameter (200–800 nm) and is relatively less electron-dense (Fig. 4F, arrowheads), whereas the other has a smaller diameter (20-40 nm) and is relatively more electron-dense (Fig. 4F, arrows). Similar electron-dense granules were also observed in LN229$^{TAZ(4SA)}$ tumor cells cocultured with d32Dcl3 neutrophils (Fig. EV3C, arrowheads and arrows). Since these granules are commonly observed in neutrophils instead of normal tumor cells (Fig. EV3D), they are likely transferred from neutrophils into tumor cells. To examine whether these granules contain MPO, we performed immune-gold TEM using an antibody against MPO. In the coculture containing LN229$^{TAZ(4SA)}$ cells and dHL-60 neutrophils, MPO-immune-gold

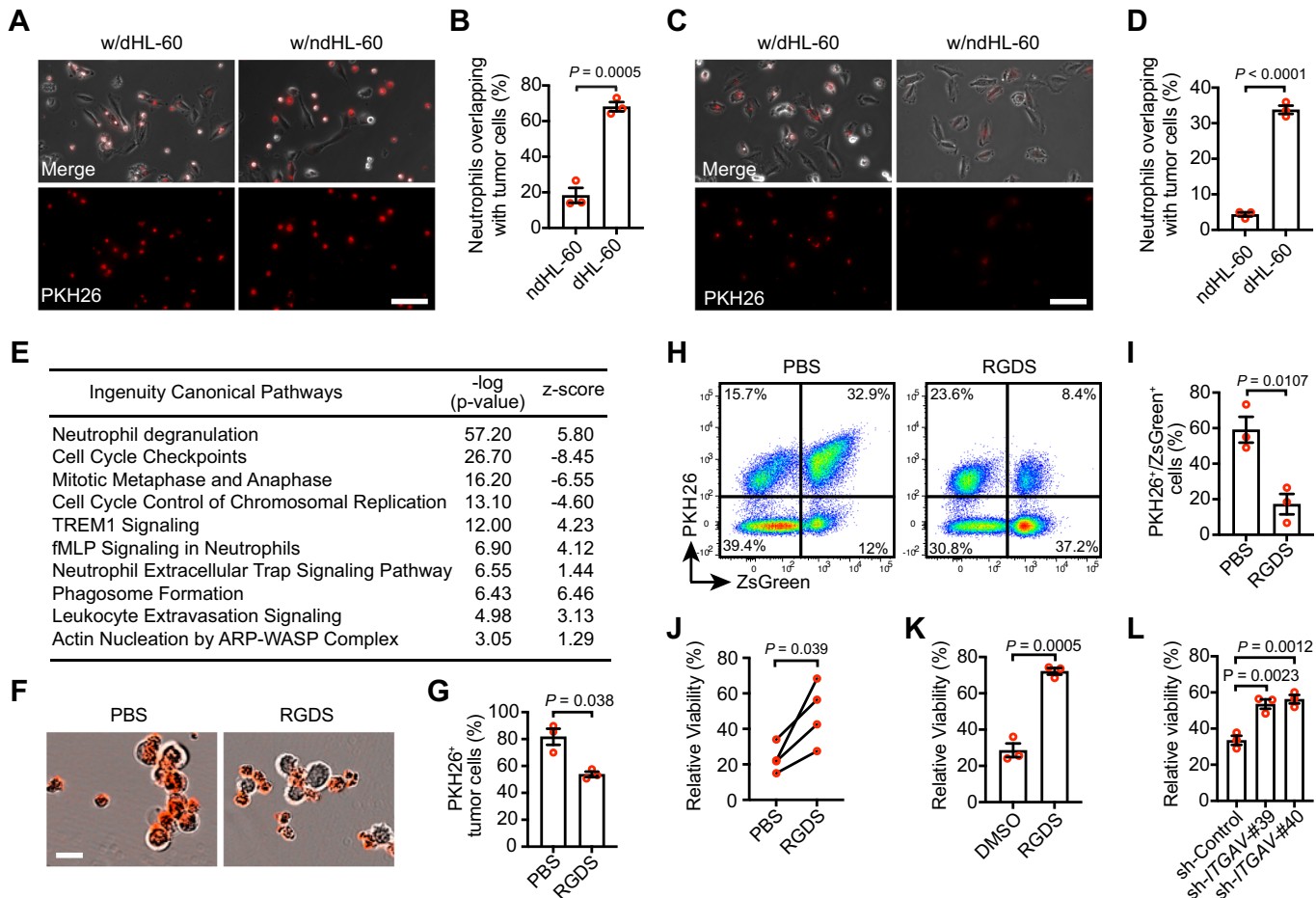

**Figure 3. Integrin-mediated cell adhesion is required for tumor cell-killing by neutrophils.**

(A) Representative images showing LN229$^{TAZ(4SA)}$ cells cocultured with PKH26-labeled dHL-60 and ndHL-60 cells ($n = 3$). Scale bar, 100 μm. (B) Quantification of LN229$^{TAZ(4SA)}$ cells with overlapping PKH26-labeled dHL-60 and ndHL-60 cells ($n = 3$ independent experiments). Unpaired t-test. (C) Representative images showing LN229$^{TAZ(4SA)}$ cells cocultured with PKH26-labeled dHL-60 and ndHL-60 cells. Wells were washed with fresh complete RPMI media, three times with agitation, before images were taken ($n = 3$). Scale bar, 100 μm. (D) Quantification of LN229$^{TAZ(4SA)}$ cells with overlapping PKH26-labeled dHL-60 and ndHL-60 cells ($n = 3$ independent experiments). Unpaired t-test. (E) Top-regulated pathways in dHL-60 cells compared to ndHL-60 cells. Right-tailed Fisher's exact test. (F) Representative images showing LN229$^{TAZ(4SA)}$ cells cocultured with PKH26-labeled dHL-60 cells with PBS or RGDS (1 mM) ($n = 3$). Scale bar, 20 μm. (G) Quantification of LN229$^{TAZ(4SA)}$ cells with overlapping PKH26-labeled dHL-60 cells ($n = 3$ independent experiments). Unpaired t-test. (H) Representative flow cytometry analysis for LN229$^{TAZ(4SA)}$ cells expressing ZsGreen cocultured with PKH26-labeled dHL-60 cells with PBS or RGDS (1 mM). Percentages for cells in each quadrant are listed ($n = 3$). (I) Quantification of cells from (H) flow cytometry analysis with both PKH26 and ZsGreen signals. The percentage of the top right quadrant was divided by the total percentage of the top right and bottom right quadrants ($n = 3$ independent experiments). Unpaired t-test. (J, K) Cancer cell viability determined using luciferase readouts for LN229$^{TAZ(4SA)}$ cells cocultured with dHL-60 cells (J) or TAN (K) with PBS or RGDS (1 mM). Luminescence readouts of cocultured wells were normalized to their respective LN229$^{TAZ(4SA)}$ cells monocultured wells. In (J), $n = 4$ independent experiments, Paired t-test; in (K), $n = 3$ independent experiments, Unpaired t-test. (L) Cancer cell viability determined using luciferase readouts for LN229$^{TAZ(4SA)}$ cells cocultured with dHL-60 cells transduced with indicated shRNAs. Luminescence readouts of cocultured wells were normalized to their respective monocultured wells for each LN229$^{TAZ(4SA)}$ cell line ($n = 3$ independent experiments). One-way ANOVA. Error bars, s.e.m. Source data are available online for this figure.

particles were found in isolated neutrophils, but not in tumor cells when they have no internalized dHL-60 cells. In those LN229$^{TAZ(4SA)}$ cells containing internalized neutrophils, we found MPO-immune-gold particles in the areas corresponding to the fragmented neutrophils (Fig. 4G). These MPO-immune-gold particles mostly associate with certain electron-dense granules, which were likely released from the fragmented neutrophil (Fig. 4G, insets). Overall, these observations indicated that neutrophil content transfer occurs through engulfment of neutrophils by tumor cells. Neutrophil-derived granules are likely released into tumor cells following such internalization.

## Statins can inhibit neutrophil internalization by tumor cells

The above results showed that several MVA metabolic pathway inhibitors are able to inhibit the neutrophil content transfer to tumor cells (Fig. 1J–L). We noticed that LN229$^{TAZ(4SA)}$ tumor cells treated by statins or GGTI-298 showed a more spindle-shaped morphology than those treated by DMSO or YM-53601 (Fig. 5A). Since statins and GGTI-298, but not YM-53601, can inhibit Rho and Rac geranylgeranylation, which is important for their activation (Fig. 1D) (Brown et al, 2006), this unique morphology

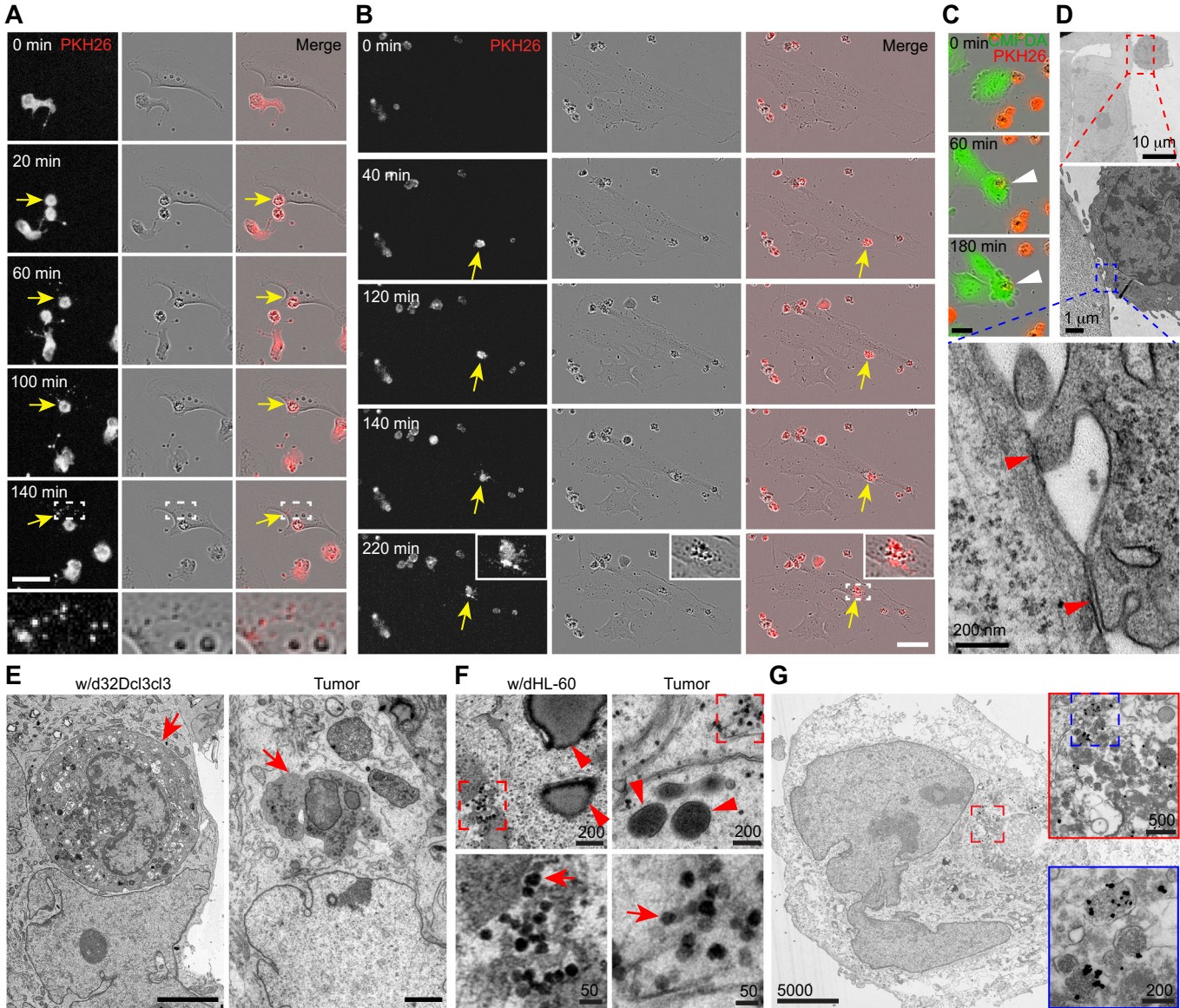

**Figure 4. Neutrophil internalization by tumor cells.**

(A, B) Time-lapse imaging of (A) LN229^TAZ(4SA) cells or (B) primary patient-derived GBM cells cocultured with PKH26-labeled d32Dcl3cl3 cells. Arrows denote the internalization of d32Dcl3cl3 cells and fragmentation of d32Dcl3cl3 contents. The outlined areas are enlarged and shown on the bottom (A) or in the insets (B). Scale bar, 50 μm. (C) Time-lapsed imaging of LN229^TAZ(4SA) cells labeled with 5-chloromethylfluorescein diacetate (CMFDA) CellTracker Green cocultured with d32Dcl3cl3 cells labeled with PKH26. Arrowheads point to a tumor cell engulfing a neutrophil and undergoing cell death. Scale bar, 20 μm. (D) Low- and high-magnification transmission electron microscopy (TEM) images of LN229^TAZ(4SA) cells cocultured with d32Dcl3cl3 cells. The outlined areas are sequentially enlarged and shown as indicated. Arrowheads point adhesion between a tumor cell and a neutrophil. (E) TEM image of LN229^TAZ(4SA) cells cocultured with d32Dcl3cl3 cells (left) or LN229^TAZ(4SA) tumor sections (right). Arrows point engulfed neutrophils. Scale bar, 5 μm (left), 2 μm (right). (F) TEM images of LN229^TAZ(4SA) cells cocultured with dHL-60 cells (left) or LN229^TAZ(4SA) tumor sections (right). The outlined areas were shown on the bottom for each. Arrowheads point larger granules. Arrows point to smaller granules. Scale bars in nm. (G) Immuno-gold TEM images of LN229^TAZ(4SA) cells cocultured with dHL-60 cells. The color-coded outlined areas were sequentially enlarged and shown in the insets. Scale bars in nm. Source data are available online for this figure.

is likely due to dysfunction of Rho or Rac signaling. To examine if Rho or Rac activity is important for neutrophil-mediated cell killing, we used Rho inhibitor I (Rho-In1) and NSC23766 trihydrochloride (NSC23766) to inhibit Rho and Rac1, respectively. Adding each of these inhibitors can markedly increase the viability of LN229^TAZ(4SA) tumor cells cocultured with dHL-60 neutrophils (Fig. 5B). Using the coculture adhesion assay, we found that

attachment of dHL-60 neutrophils to LN229^TAZ(4SA) tumor cells was not affected in the presence of simvastatin, fluvastatin, GGTI-298, or YM-53601 compared to the DMSO control (Fig. EV4A,B). Neutrophils attaching to tumor cells can be classified into three tiers, including attached, entering, and internalized (Fig. 5C,D). When examining the three tiers of cells in the presence of each inhibitor, we found that administration of simvastatin, fluvastatin,

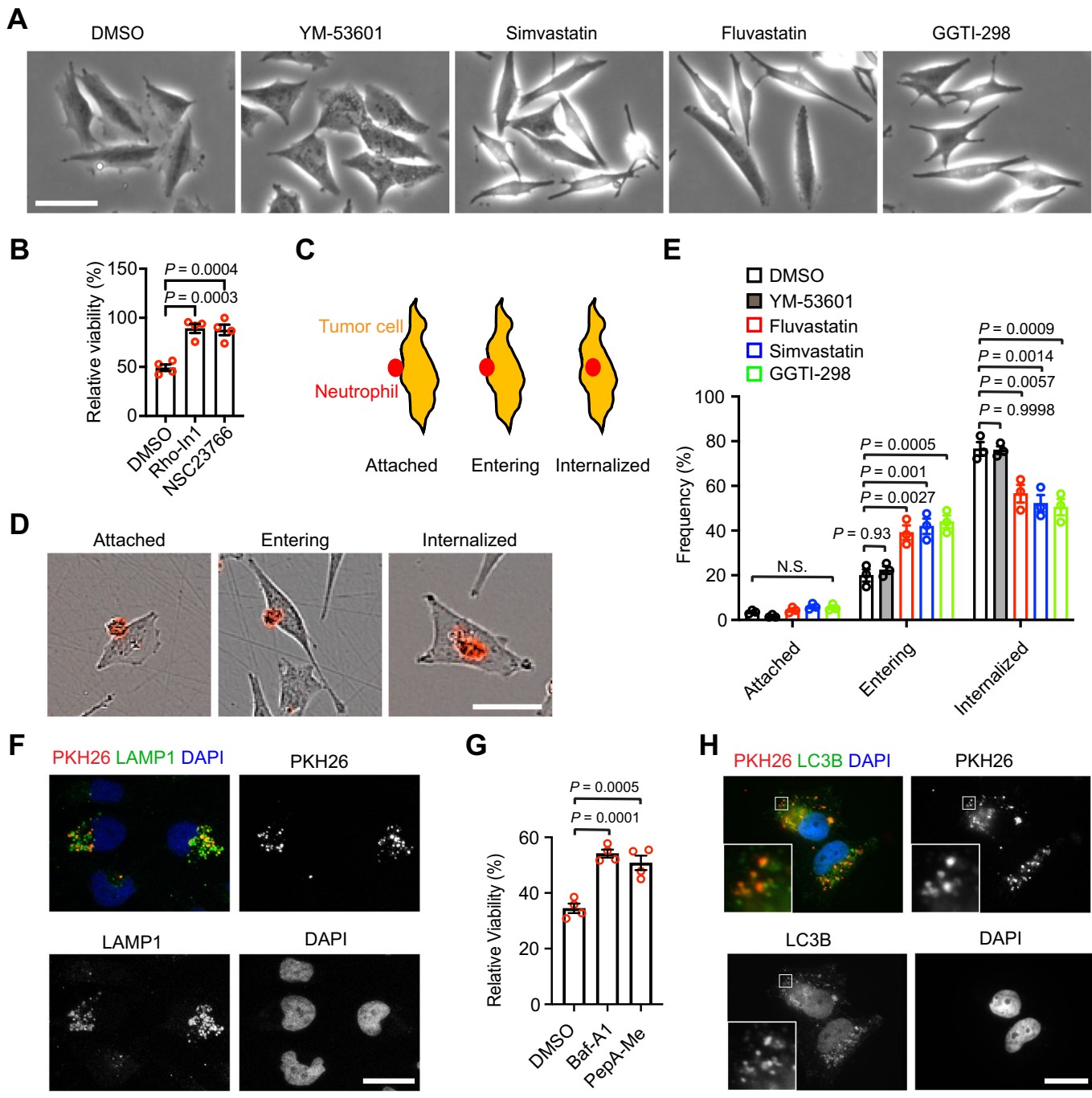

**Figure 5. Statins can inhibit the neutrophil internalization by tumor cells.**

(A) Representative images showing the morphology of LN229^TAZ(4SA) cells after 12 h treatment with DMSO, simvastatin (2.5 μM), fluvastatin (2.5 μM), GGTI-298 (10 μM), or YM-53601 (10 μM). Scale bar, 50 μm. (B) Luciferase assay results for LN229^TAZ(4SA) cells cocultured with dHL-60 cells with Rho inhibitor 1 (Rho-In1) (500 nM) or NSC23766 (100 μM) (n = 4 independent experiments). Luminescence readouts of cocultured wells were normalized to their respective LN229^TAZ(4SA) cells monocultured wells, both with drugs. One-way ANOVA. (C) Depictions of different tiers of neutrophil (red oval) internalization by LN229^TAZ(4SA) tumor cells: attached (Tier 1), entering (Tier 2), and internalized (Tier 3). (D) Representative images showing the internalization tiers depicted in (C). Scale bar, 50 μm. (E) Quantification of frequency of dHL-60 internalization tiers during coculture with LN229^TAZ(4SA) cells in the conditions described in (A). ~300 cells surveyed per repeat (n = 3 independent experiments). One-way ANOVA. (F) Representative immunofluorescent images showing LAMP1 staining, PKH26 signal, and DAPI staining for LN229^TAZ(4SA) cells cocultured with PKH26-labeled dHL-60 cells (n = 3). Scale bar, 20 μm. (G) Cancer cell viability determined using luciferase readouts for LN229^TAZ(4SA) cells cocultured with dHL-60 cells with DMSO, Bafilomycin A1 (400 nM), or pepstatin A-methyl ester (25 μM). Luminescence readouts of cocultured wells were normalized to their respective LN229^TAZ(4SA) cells monocultured wells, both with indicated drugs (n = 4 independent experiments), one-way ANOVA. (H) Representative immunofluorescent images showing LC3B staining, PKH26 signal, and DAPI staining for LN229^TAZ(4SA) cells cocultured with PKH26-labeled dHL-60 cells (n = 3). The outlined areas are enlarged and shown in the insets. Scale bar, 20 μm. Error bars, s.e.m. Source data are available online for this figure.

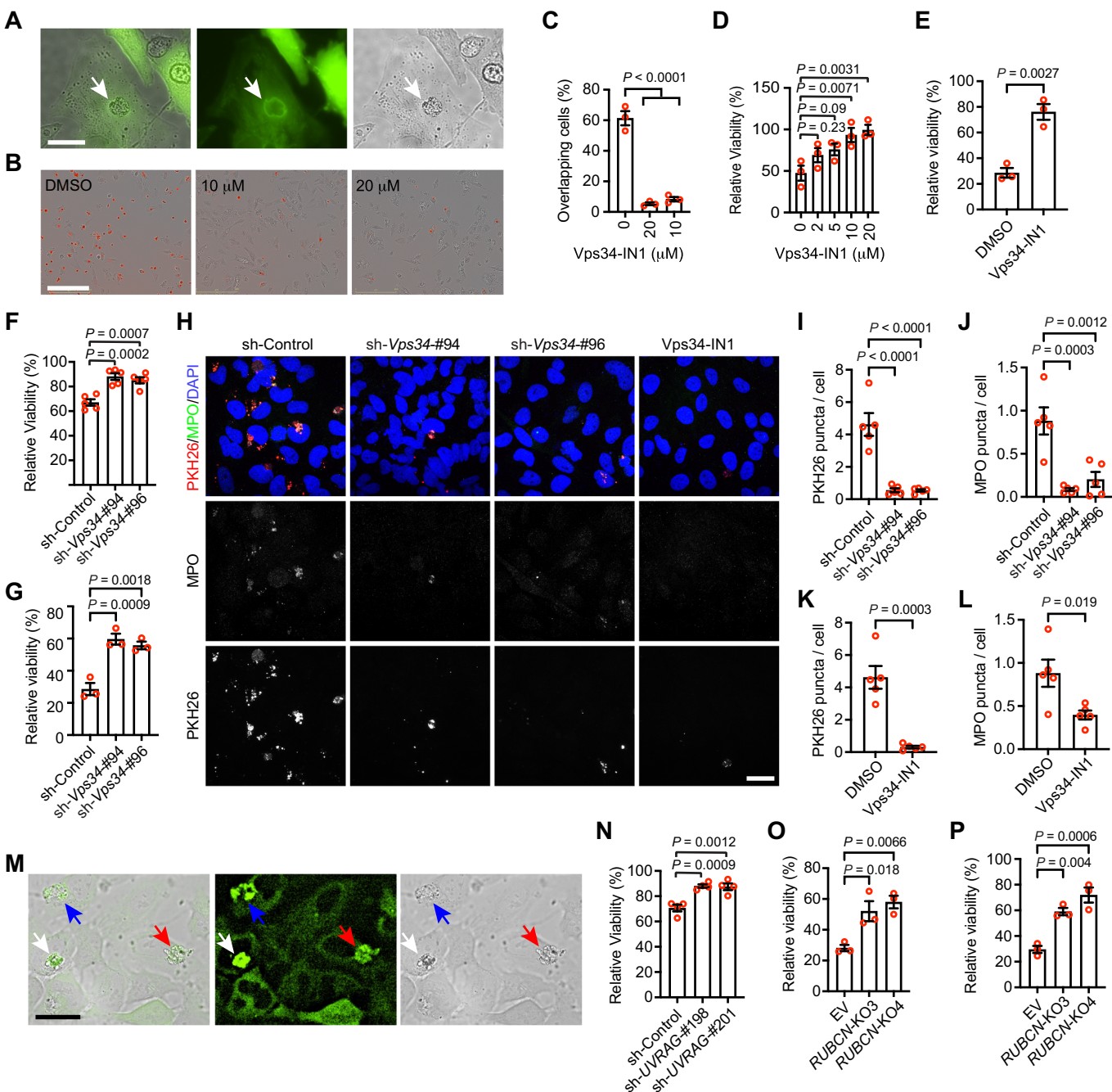

or GGTI-298, but not YM-53601, increased the population frequency of the entering tier and decreased that of the internalized tier (Fig. 5E). These results suggested that simvastatin, fluvastatin, and GGTI-298 suppressed neutrophil content transfer through inhibition of the internalization process.

## LC3-associated phagocytosis (LAP) is responsible for neutrophil internalization by tumor cells

To trace the fate of neutrophils in tumor cells after internalization, we used lysosomal-associated membrane protein 1 (LAMP1) as a marker for lysosomes. Immunofluorescence imaging showed that the dHL-60

or d32Dcl3 neutrophil fragments in LN229$^{TAZ(4SA)}$ tumor cells colocalize with LAMP1 (Figs. 5F and EV4C), suggesting they undergo lysosome-mediated degradation following internalization by tumor cells. To examine if functional lysosomes are required for tumor cell death, we used bafilomycin A1 (a vacuolar H$^+$-ATPase inhibitor) or pepstatin A-methyl ester (PepA-Me, an aspartyl protease inhibitor) to inhibit lysosomal functions (Mai et al, 2017; Torii et al, 2016). In the presence of either agent, dHL-60 neutrophil-induced tumor cell death was inhibited (Fig. 5G), suggesting that lysosomes are involved in the tumor cell death induction by neutrophils.

Neutrophil internalization into tumor cells followed by lysosome-mediated fragmentation mimics other cell engulfment

**Figure 6.   LC3-associated phagocytosis is responsible for neutrophil internalization by tumor cells.**

(A) Live-cell imaging of LN229$^{TAZ(4SA)}$ cells stably expressing GFP-LC3 cocultured with dHL-60 cells. 1 h coculture time. Arrows denote GFP-LC3 rings forming around internalized HL-60 cells. Scale bar, 20 μm. (B) Representative images showing LN229$^{TAZ(4SA)}$ cells cocultured with PKH26-labeled dHL-60 cells with DMSO or Vps34-IN1. Wells were washed three times with complete RPMI media before images were taken ($n = 3$). Scale bar, 200 μm. (C) Quantification of LN229$^{TAZ(4SA)}$ cells with overlapping PKH26-labeled dHL-60 cells after coculture in DMSO or Vps34-IN1 ($n = 3$ independent experiments). One-way ANOVA. (D, E) Cancer cell viability determined using luciferase readouts for LN229$^{TAZ(4SA)}$ cells cocultured with dHL-60 cells (D) or TAN (E) with DMSO or Vps34-IN1 (10 μM in (E)). Luminescence readouts of cocultured wells were normalized to their respective LN229$^{TAZ(4SA)}$ cells monocultured wells, both with indicated drugs ($n = 3$ independent experiments), one-way ANOVA. (F, G) Cancer cell viability determined using luciferase readouts for LN229$^{TAZ(4SA)}$ cells transduced with indicated shRNAs cocultured with dHL-60 cells (F, $n = 5$ independent experiments) or TAN (G, $n = 3$ independent experiments). Luminescence readouts of cocultured wells were normalized to their respective monocultured wells for each LN229$^{TAZ(4SA)}$ cell line. One-way ANOVA. (H) Representative images showing immunofluorescent PKH26, MPO, and DAPI signal for LN229$^{TAZ(4SA)}$ cells cocultured with PKH26-labeled TAN ($n = 3$ tumor-bearing animals). LN229$^{TAZ(4SA)}$ cells were transduced with indicated shRNAs, or the coculture was treated by Vps34-IN1 (10 μM). Scale bar, 10 μm. (I–L) Quantification of PKH26$^+$ puncta or MPO$^+$ puncta in each tumor cells in the experiments as shown in (H). Two fields of view were used for each group in each experiment ($n = 5$ randomly picked microscopy fields). One-way ANOVA. (M) Live-cell imaging of LN229$^{TAZ(4SA)}$ cells stably expressing UVRAG-GFP cocultured with dHL-60 cells. 1 h coculture time. Arrows denote elevated UVRAG-GFP signals colocalizing with internalized dHL-60 cells. Scale bar, 20 μm. (N–P) Cancer cell viability determined using luciferase readouts for LN229$^{TAZ(4SA)}$ cells transduced with indicated shRNAs (N) or gRNA (O, P) cocultured with dHL-60 cells (N, O) or TAN (P). Luminescence readouts of cocultured wells were normalized to their respective monocultured wells for each LN229$^{TAZ(4SA)}$ cell line ($n = 4$ independent experiments in (N), $n = 3$ independent experiments in (O) and (P)). One-way ANOVA. Error bars, s.e.m. Source data are available online for this figure.

phenomena, such as entosis and efferocytosis (Boada-Romero et al, 2020; Krishna and Overholtzer, 2016). Both cellular processes involve microtubule-associated protein 1 A/1B light chain 3 (LC3), therefore belonging to LC3-associated phagocytosis (LAP). To examine if neutrophil internalization into tumor cells also belongs to LAP, we first used the antibody recognizing LC3 isoform B (LC3B) and found that the dHL-60 neutrophil fragments are reactive to the LC3B antibody (Fig. 5H). We then employed LN229 cells stably expressing GFP-LC3. When dHL-60 neutrophils were cocultured with these tumor cells, we found neutrophils were enclosed by a GFP ring-like structure when being internalized into the tumor cells (Fig. 6A, arrows). In LAP, the PI3K complex containing Vps34, Beclin1, UVRAG, Vps15, and Rubicon is responsible for activating the LC3 ligation machinery (Boada-Romero et al, 2020). To examine if the complex is required for the engulfment of neutrophils by tumor cells and subsequent tumor cell death, we performed loss of function studies of Vps34, the catalytic subunit of the PI3K complex. Tumor cells cultured in the presence of Vps34-IN1, a selective inhibitor of Vps34, showed dosage-dependent accumulation of cellular vacuoles (Fig. EV5A), suggesting cellular vesicle shuttling is disrupted when Vps34 is inhibited. In the presence of Vps34-IN1, dHL-60 neutrophil engulfment by tumor cells was markedly inhibited (Fig. 6B,C). In addition, dHL-60 neutrophil-induced tumor cell death was also inhibited in a dosage-dependent manner (Fig. 6D). TAN-induced LN229$^{TAZ(4SA)}$ tumor cell-killing was also inhibited by Vps34-IN1 (Fig. 6E). To further examine the role of Vps34, we used two different shRNAs to knock down Vps34 expression in LN229$^{TAZ(4SA)}$ cells (Fig. EV5B). Depletion of Vps34 also leads to accumulation of cellular vacuoles (Fig. EV5C), though not to the extent when compared to treatment with Vps34-IN1. We found that *Vps34* depletion in LN229$^{TAZ(4SA)}$ cells also reduced the engulfment of dHL-60 neutrophils by tumor cells (Fig. EV5D,E) and tumor cell-killing by the neutrophils (Fig. 6F). Depletion of *Vps34* in LN229$^{TAZ(4SA)}$ cells also rescue them from TAN-mediated cell killing (Fig. 6G). To examine whether neutrophilic content transfer to tumor cells also requires Vps34, we pre-labeled TAN with PKH26 and cocultured them with LN229$^{TAZ(4SA)}$ cells. When Vps34 expression in LN229$^{TAZ(4SA)}$ cells was silenced by shRNAs, or its activity was inhibited by Vps34-IN1, PKH26+ and MPO+ granules in tumor cells are markedly fewer than controls

(Fig. 6H–L). UVRAG is another key component of the PI3K complex. When LN229 cells expressing GFP-tagged UVRAG (UVRAG-GFP) were cocultured with dHL-60 cells, we found that UVRAG-GFP accumulated along with the engulfed neutrophils (Fig. 6M, arrows). To examine the role of UVRAG, we silenced its expression in LN229$^{TAZ(4SA)}$ cells with two different shRNAs (Fig. EV5F). Like *Vps34*, *UVRAG* depletion can also induce accumulation of cellular vacuoles (Fig. EV5G) and reduce tumor cell death induced by dHL-60 neutrophils (Fig. 6N). *RUBCN* is a gene essential for LAP, but not several other cellular membrane-remodeling processes, such as autophagy and endocytosis (Galluzzi and Green, 2019). Depletion of *RUBCN* through CRISPR in LN229$^{TAZ(4SA)}$ cells (Fig. EV5H) increased their survival when being cocultured with dHL-60 or TAN (Fig. 6O,P). Overall, the above results indicated that LAP activated by the Vps34-UVRAG-RUBCN-containing PI3K complex is responsible for neutrophil internalization by tumor cells and thereafter required for neutrophil-induced tumor cell death.

## Myeloperoxidase inhibition or Vps34 depletion reduces GBM necrosis and aggressiveness

Previous studies found that myeloperoxidase (MPO) is involved in promoting tumor cell ferroptosis in vitro (Yee et al, 2020). Consistently, results in the current studies also found a strong association between the transfer of MPO-containing granules into tumor cells and tumor cell death. To examine if MPO is involved in promoting necrosis development in GBM, we used 4-aminobenzoic acid hydrazide (4-ABAH), a small-molecule inhibitor which can specifically inhibit the peroxidase activity of MPO (Kettle et al, 1995; Ruggeri et al, 2015). Mice bearing LN229$^{TAZ(4SA)}$ tumors were treated with either 4-ABAH or a vehicle control. 4-ABAH-treated mice lived significantly longer (12.5%) than controls (Fig. 7A). Histological evaluation of similar-sized tumors from the treated mice indicated that the necrosis in 4-ABAH-treated tumors was significantly smaller (33%) than in the control tumors (Fig. 7B,C). These results suggested that MPO enzymatic activity is involved in necrosis development and tumor progression in GBM.

It was shown that inhibition of ferroptosis by ectopically expressing GPX4 or silencing the expression of ACSL4 can reduce necrosis by 47% or 58–68%, respectively, and prolong the survival

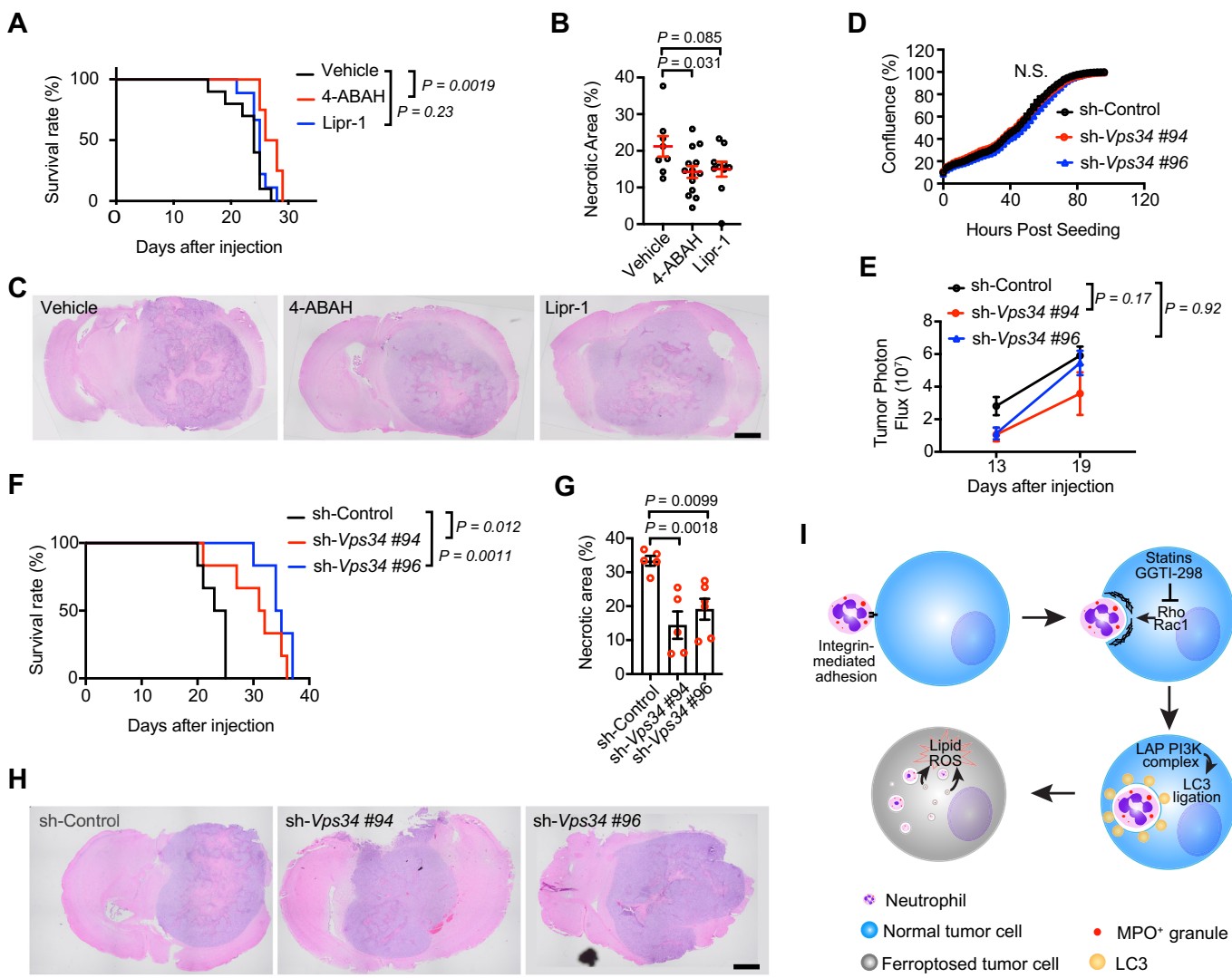

**Figure 7. Myeloperoxidase inhibition or Vps34 depletion reduces GBM necrosis and aggressiveness.**

(A) Kaplan–Meier survival curve for mice implanted with LN229$^{TAZ(4SA)}$ tumors treated by indicated chemicals ($n_{Vehicle} = 10$, $n_{4-ABAH} = 8$, $n_{Lipr-1} = 9$ mice for each group). Log-rank test. (B) Tumor necrotic area percentage was calculated by dividing the necrotic area size by total tumor area size. Each data point represents one mouse. Unpaired t-test. (C) Representative H&E brain sections at endpoints for each mice group with LN229$^{TAZ(4SA)}$ tumors treated by indicated chemicals. Scale bar, 1000 μm. (D) Growth curve of LN229$^{TAZ(4SA)}$ cells transduced by indicated shRNAs measured by live-cell time-lapse imaging ($n = 4$). One-way ANOVA. (E) Size of tumors derived from LN229$^{TAZ(4SA)}$ cells transduced by indicated shRNAs was monitored using bioluminescence via IVIS imaging system. Photon emissions were quantified on days 13 and 19 after tumor cell injection ($n = 6$ mice for each group). Two-Way ANOVA. (F) Kaplan–Meier survival curve for mice implanted with LN229$^{TAZ(4SA)}$ cells transduced by indicated shRNAs ($n = 6$ mice for each group). Log-rank test. (G) Tumor necrotic area percentage was calculated by dividing the necrotic area size by the total tumor area size. Each data point represents one tumor from one mouse. One-way ANOVA. (H) Representative H&E brain sections at endpoints for each mice group with tumors derived from LN229$^{TAZ(4SA)}$ cells transduced by indicated shRNAs. Scale bar, 1000 μm. (I) Working model depicting the role of LAP and other components in neutrophil-induced cancer cell death. Terminal differentiated neutrophils increase their adhesion ability to tumor cells through the expression of cell adhesion genes, such as integrins. Subsequently, adhesion of neutrophils triggered their engulfment by tumor cells through LAP. The engulfment can be inhibited by statins or a geranylgeranylation inhibitor, which may inhibit Rac1- or Rho-mediated actin cytoskeleton remodeling. Following the engulfment, neutrophils are fragmented through a lysosomal-mediated process and then induce tumor cell ferroptosis through releasing their contents, such as MPO. Error bars, s.e.m. Source data are available online for this figure.

of tumor-bearing mice (Yee et al, 2020). To examine if inhibiting ferroptosis pharmacologically can also reduce necrosis development and prolong survival, we used liproxstatin-1, a ferroptosis inhibitor that is likely a radical-trapping antioxidant (Friedmann Angeli et al, 2014; Zilka et al, 2017). The size of necrosis in liproxstatin-1-treated tumors did not show significant reduction than in the control tumors (Fig. 7B), suggesting the treatment is less effective in suppressing ferroptosis in the tumors than the genetic

manipulations. Under this situation, Treatment with liproxstatin-1 did not improve the survival of tumor-bearing mice (Fig. 7A).

With the above finding that LAP is involved in neutrophil-induced tumor cell death, we examined if blocking LAP could also inhibit necrosis formation in LN229$^{TAZ(4SA)}$ tumors. Silencing *Vps34* expression in LN229$^{TAZ(4SA)}$ cells did not affect cell proliferation in vitro (Fig. 7D). When examining the tumor growth at an early tumor development stage (up to day 19 after tumor implantation

(ATI)) through bioluminescence imaging, we found that *Vps34*-depleted tumors did not show significant difference compared to control tumors (Fig. 7E). Notably, necrosis is much less prominent (especially up to day 16 ATI) than that after day 20 ATI (Yee et al, 2022). Therefore, evaluating the effect of *Vps34*-depletion on tumor necrosis at this early stage is less feasible than later stages. However, when examining the mice survival, we found that mice implanted with *Vps34*-depleted tumor cells lived significantly longer (31% and 44%, for sh#94 and sh#96, respectively) than controls (Fig. 7F). Through examining similar-sized tumors, we found that necrosis in *Vps34*-depleted tumors was significantly smaller (57% and 43%, for sh#94 and sh#96, respectively) than in the control tumors (Fig. 7G,H). These results suggested that LAP is important for tumor necrosis development and that blocking LAP through inhibiting Vps34 can reduce tumor aggressiveness. Overall, these results supported that blocking necrosis formation through harnessing neutrophils-induced tumor cell death could reduce the aggressiveness of GBM.

## Discussion

In this study, we elucidated a surprising mechanism through which neutrophils induce tumor cell death in the development of GBM necrosis. This mechanism involves the transfer of neutrophilic contents into tumor cells following the complete engulfment of neutrophils by tumor cells. Upon terminal differentiation, neutrophils enhance their adhesion ability to tumor cells by upregulating the expression of cell adhesion genes, such as integrins. Subsequently, the adhesion of neutrophils triggers their engulfment by tumor cells through LAP. Following engulfment, neutrophils undergo fragmentation via a lysosome-mediated process and subsequently induce tumor cell death by releasing their contents, such as MPO. In combination with the previous report (Yee et al, 2020), our studies suggested LAP-mediated neutrophil internalization is responsible for the neutrophilic contents transfer to tumor cells and subsequently inducing the later cell ferroptosis (Fig. 7I).

Previous studies have indicated that the cytotoxicity of neutrophils against tumor cells in most situations relies on an oxidative process (Clark and Klebanoff, 1975, 1979) (Clark and Szot, 1981) (Slivka et al, 1980; Weiss and Slivka, 1982). This process requires direct contact, which can trigger neutrophils to secrete $H_2O_2$ (Granot et al, 2011) (Saito et al, 1992). Our results are consistent with these previous observations, underscoring the necessity of direct cell contact for neutrophil-mediated tumor cell killing. Interestingly, our results suggest that while contact is necessary, its alone is insufficient; engulfment by tumor cells also appears to be required for effective cell killing. It would be interesting to know whether a similar engulfment process contributes to neutrophil-induced cell killing in other contexts.

LAP is a process that utilizes autophagy machinery to clean up dead or dying cells by myeloid cells through efferocytosis (Boada-Romero et al, 2020). It also occurs when living cells invade their neighboring cells, known as entosis (Krishna and Overholtzer, 2016). Here, we describe a third type of cell engulfment belonging to LAP. In this situation, terminally differentiated neutrophils are engulfed by tumor cells. Through this process, the engulfed neutrophil, by sacrificing itself, initiates an oxidative cell-killing mechanism that induces tumor

cell ferroptosis. In GBM, this phenomenon contributes to the enlargement of necrotic areas, potentially fueling tumor progression (Yee and Li, 2021; Yee et al, 2020).

Our unbiased screening revealed that statins can inhibit neutrophil-mediated tumor cell-killing by blocking the neutrophil internalization process. By dissecting the downstream effector pathways controlled by the mevalonate signaling, we identified the protein geranylgeranylation pathway as a key player in this internalization process. Geranylgeranylation of Rho and Rac proteins is crucial for their activation and can be inhibited by statins (Brown et al, 2006). Morphologic changes observed in response to treatments with simvastatin, fluvastatin, or GGTI-298, are consistent with the inactivation of Rho and Rac. By employing inhibitors of Rho or Rac1, we demonstrated that neutrophil-mediated tumor killing is inhibited. Given that Rho and Rac1 are important for remodeling the actin cytoskeleton, it is likely that statins inhibit LAP-mediated neutrophil internalization by impeding the actin cytoskeleton remodeling required for the cell engulfment.

As a hallmark of GBM, necrosis is thought to contribute to tumor progression and resistance to therapies. Therefore, finding a way to inhibit necrosis is desirable. In this study, we explored three approaches to harness neutrophil-induced tumor cell ferroptosis. Initially, we investigated the inhibition of myeloperoxidase (MPO) using its inhibitor, 4-ABAH, which significantly reduced necrosis in GBM and extended the survival of tumor-bearing mice. However, our second approach, utilizing liproxstatin-1 to block tumor cell ferroptosis, showed limited efficacy in slowing down tumor necrosis development, likely due to its insufficient potency in vivo at the tested dosage. Treatment with liproxstatin-1 did not confer a survival benefit to mice, possibly due to its inability to adequately suppress necrosis or unidentified side effects on the treated mice. In our third approach, we targeted LAP through Vps34 depletion. While this intervention did not significantly impede tumor growth during the early stages of tumor development, it effectively inhibited necrosis development and led to prolonged mouse survival. It is important to note that in vivo tumorigenesis experiments inherently entail various influencing factors, warranting cautious interpretation of the results presented here. Nonetheless, these findings suggest that inhibiting necrosis may hold promise in improving outcomes for GBM patients by mitigating tumor aggressiveness.

## Methods

### Isolation and generation of patient-derived primary GBM tumor cells

All experiments with patient-derived material were performed in accordance with the guidelines of the Biorepository Committee at the Penn State Hershey Neuroscience Institute (PSHNI). All human materials utilized in this study were de-identified specimens and exempt from review by Penn State College of Medicine Institutional Review Board. Tissue collection and generation of primary tumor cells were detailed as follows. Briefly, pieces of tumor tissues were collected from patients diagnosed with brain malignancy based on radiographic imaging undergoing craniotomy at the Department of Neurosurgery, Penn State Hershey Medical Center, and distributed

by the PSHNI Biorepository Committee staff after de-identification. Among above, only patients with preliminary intra-op pathological reports confirmed high-grade gliomas were collected. Final biopsy-proven tissue-based diagnoses of grade VI high-grade gliomas (a.k.a. GBMs) were distributed by the PSHNI Biorepository committee once the pathology reports were finalized and again de-identified. Tumor tissue was minced into small pieces (no larger than 1 mm in diameter), followed by washing with cold 1X DPBS three times (10 mL each time) to remove blood and debris as much as possible. Minced and washed tumor pieces were then incubated with 10 mL of tumor digestion medium containing 1 μg/μL hyaluronidase and 1 mg/mL collagenase IV at 37 °C for 15 min. Following digestion, tissue pieces were physically dissociated first using 5 ml serological pipettes and then with P1000 pipette tips (by forcefully pipetting up and down approximately 20 times) until chunks of tissue were no longer visible and the solution appeared homogeneously cloudy. Dissociated cell mixtures were then filtered with a cell strainer (mesh pore diameter 40 μm) into a 50 mL Falcon conical tube, pelleted via centrifugation at 1000 rpm for 5 min ($110 \times g$; ST-40 centrifuge, ThermoFisher) to remove the digestion medium, and washed three times with F12/DMEM medium. Leftover pieces of tumor tissues on the strainer were also included and cultured in vitro. Patient-derived primary GBM tumor cells were cultured in DMEM supplemented with 10% fetal bovine serum and 1% Antibiotic–Antimycotic Solution at 37 °C with 5% $CO_2$. Full culture medium consists of Dulbecco's modified Eagle's medium (DMEM; 10-013-CV, Corning), 10% fetal bovine serum (FBS; Gibco, 10437028), and 1% Antibiotic–Antimycotic Solution (stock is 100X, 30-002-Cl, Corning). Hyaluronidase stock powder (100 mg) was reconstituted in 0.5 mL $Ca^{2+}$ free and $Mg^{2+}$ free HBSS (0.2 g/mL, stock is 200X) and collagenase IV stock powder (1 g) was reconstituted in 10 mL $Ca^{2+}$ free and $Mg^{2+}$ free HBSS (100 mg/mL, stock is 100X), and both stocks were stored at −20 °C. Tumor tissue digestion medium consists of F12/DMEM (without L-glutamine; Corning 15-090-CV), 1X of hyaluronidase as well as collagenase IV.

## Mice and orthotopic xenograft tumor models

The in vivo experiments were performed following the protocol detailed in (Yee et al, 2020). In brief, 6–8-week-old female athymic nude mice (Nu(NCr)-Foxn1nu Strain Code: 490, Charles River) were used as the GBM mouse model. For each mouse, $3 \times 10^5$ indicated human GBM cells expressing firefly luciferase were injected into the frontal right hemisphere at coordinates ($+1, +2, -3$). Tumor size was monitored using bioluminescence via IVIS imaging system (Xenogen, Alameda, CA) and photon emissions were quantified with Living-Image software (Xenogen). For 4-ABAH and Liproxstatin-1 treatments, bioluminescence imaging (BLI) was performed two weeks post-implantation to verify the location of tumor growth. Mice were then randomly assigned into three treatment groups based on their BLI readings, with each group receiving vehicle control (2% DMSO, 40% PEG300, 2% Tween-80 in PBS), 4-ABAH (twice daily at 20 mg/kg), or Liproxstatin-1 (once daily at 10 mg/kg), respectively, through intraperitoneal injection. To examine necrosis, tumors in each group were collected from mice brains whenever vehicle-treated mice reach the terminal stage. Mouse body weight was monitored in each group as a benchmark for tumor procession (above 25% body weight loss). Whole brains were collected and fixed using 4% formalin for 2 days,

and submitted to Penn State College of Medicine's Comparative Medicine Histology Core for processing. The brain samples were cut into 5-μm-thick sections, and stained with hematoxylin & eosin (H&E). For quantification of tumor necrosis, the entire tumor area and necrotic zones were manually drawn and measured using ImageJ. The necrotic percentage is the necrotic area divided by the total tumor area. All mice were housed in a 12-h light/dark cycle, 18–23 °C ambient temperature, 40–60% humidity rooms with free access to standard rodent diet and water. All experiments conducted in this study were first approved by the Penn State University Institutional Animal Care and Use Committee.

## Cells

HL-60 human promyelocytic leukemia cell line (CCL-240), 32Dcl3cl3 murine myeloblastic cell line (CR-11346), and human LN229 glioblastoma cell line (CRL-2611) were purchased from ATCC. LN229 cells were cultured in Dulbecco's modified Eagle's medium (DMEM; 10-013-CV, Corning) supplemented with 10% fetal bovine serum (FBS; Gibco, 10437028) and 1% antibiotic–antimycotic Solution (30-004-CI, Corning) at 37 °C with 5% $CO_2$. Both 32Dcl3cl3 and HL-60 cells were cultured in Roswell Park Memorial Institute (RPMI) 1640 (10-040-CV, Corning) supplemented with 10% FBS and 1% antibiotic–antimycotic solution at 37 °C with 5% $CO_2$. In addition, the 32Dcl3cl3 media was supplemented with 5 ng/mL mouse interleukin-3 (IL3) (CB40040, ThermoFisher). The protocol from (Guchhait et al, 2003) was used to differentiate 32Dcl3cl3 cells into neutrophils. In brief, $3 \times 10^5$/ml cells were seeded in 100 mm cell culture dishes containing the RPMI listed above, further supplemented with 100 ng/ml of recombinant human granulocyte colony-stimulating factor (rhG-CSF; 214-CS, R&D) and cultured for 24 days. The protocol from (Breitman et al, 1980; Bunce et al, 1994; Collins, 1987) was used to differentiate HL-60 cells. In short, $3 \times 10^5$/ml cells were seeded in 100 mm cell culture dishes containing the RPMI listed above with 1 μM all-*trans*-retinoic acid (ATRA; R2625, Sigma), and 30 nM rhG-CSF for 24 days. Fresh media (3 mL) were added to 32Dcl3cl3 and HL-60 dishes every 4 days to maintain cell density of ~$3 \times 10^5$/ml cells. After 24 days, the differentiation status of the cells was confirmed using Ly6G for 32Dcl3cl3 cells and CD66b for HL-60 cells via flow cytometry. None of these cell lines were listed in the database of misidentified cell lines maintained by ICLAC and NCBI Biosample. These cell lines were not authenticated in this study. All cell lines were confirmed as *Mycoplasma* negative before experiments. For coculture experiments, cells were cultured in neutrophil RPMI media without differentiation agents.

## Antibodies, reagents, compounds

Antibodies or dyes for immunofluorescent staining: Myeloperoxidase (ab9535, Abcam), LC3B (3868, Cell Signaling Technologies), LAMP-1 (H4A3, Developmental Studies Hybridoma Lab), BODIPY 581/591 C11 (D3861, ThermoFisher Scientific), Sytox-Blue (S34857, Thermo-Fisher). Antibodies for Immunoblotting: β-Actin (A5316, Sigma-Aldrich), Vps34 (PI3 Kinase Class III, 4263, Cell Signaling Technologies), UVRAG (13115, Cell Signaling Technologies). ITGA5 (4705, Cell Signaling Technologies), ITGB1 (34971, Cell Signaling Technologies). Compounds used to treat cells: Prestwick Chemical FDA Library (provided by the Drug Discovery, Development and

Delivery core of PSU College of Medicine), simvastatin (10010344, Cayman Chemicals), fluvastatin (10010334, Cayman Chemicals), GGTI-298 (16176, Cayman Chemicals), YM-53601 (18113, Cayman Chemicals), FTI-277 (S7465, SelleckChem), RGDS (A9041, Sigma-Aldrich), Vps34-IN1 (17392, Cayman Chemicals), Liproxstatin-1 (SML1414, Sigma), 4-Aminobenzoic hydrazide (4-ABHA, 103200050, Acros Organics), IKE (S8877, SelleckChem), RSL3 (19288, Cayman Chemicals). Ferrostatin-1 (SML0583, Sigma), Liproxstatin-1 (SML1414, Sigma), Deferoxamine mesylate salt (D9533, Sigma), PF06282999 (PZ-0375 5MG, Sigma).

## Cell viability assays

LN229 cells stably expressing TAZ[4SA] and luciferase were seeded in 96-well flat-bottom plates at a density of 5000 cells/well alone or cocultured with neutrophils in a 1-to-25 ratio in duplicates and incubated at 37 °C with 5% $CO_2$ for 24 h. Next, luminescence readouts were used to determine GBM cell viability via the Luciferase Assay System (E1500, Promega) following the manufacturer's protocol. In brief, cells were washed once with DPBS and then lysed at room temperature for 2 min on a shaker. Cells were then given D-firefly luciferin potassium salt solution, and luminescence was immediately measured via a multi-mode plate reader (BMG Labtech). Blank-corrected luminescence values from cocultured wells were normalized to their respective GBM monoculture wells to calculate the relative viability. Monocultures and cocultures may be done in the presence of indicated drugs, as detailed in the figure legends. Neutrophils used include 32Dcl3cl3, HL-60, and tumor-associated neutrophils, as indicated in figure legends. Tumor cells used include genetically silenced cells for genes indicated in figure legends.

## Cell death assays

LN229[TAZ(4SA)] cells were seeded at 5000 cells/well in 96-well flat-bottom plates and cultured overnight at 37 °C with 5% $CO_2$. Appropriate drugs were added individually or in combination as detailed in figure legends. To measure cell death, 100 nM YOYO™-3 Iodide (612/631) (Y3606, Invitrogen) stain was used to assess membrane permeability via Incucyte S3 Live Cell Imaging System (Sartorius) using phase contrast and red channel. Cell death was calculated through dividing red-stained cells by phase contrast area using the Incucyte Cell-by-Cell Analysis Software Module (Sartorius). To determine relative cell death, combination treatment readouts were normalized to either RSL3 + DMSO or IKE + DMSO readouts.

## Adherence assays

For adherence assays, $2.5 \times 10^4$ LN229[TAZ(4SA)] cells were seeded in 24-well flat-bottom plates and incubated at 37 °C with 5% $CO_2$ overnight. Where appropriate, cells were pretreated with drugs as detailed in figure legends. Next, $2.5 \times 10^5$ HL-60 or 32Dcl3cl3 differentiated neutrophils, pre-labeled with PKH26 red fluorescent dye (MINI26, Sigma-Aldrich), were cocultured with the GBM cells for 1 h at 37 °C with 5% $CO_2$. Following coculture, cells were washed three times with fresh culture media and micrographs were taken under phase contrast and red IF channel. Overlapping frequency was calculated by dividing the number of tumor cells

with PKH26-stained neutrophils by total tumor cells. Attached, entering, and internalized frequencies were calculated by dividing the events in each respective category by the total sum of all three events. For RGDS adherence assays, GBM cells were detached using Accutase (00-4555-56, Invitrogen), centrifuged for 4 min at 1000 rpm (110 × *g*; ST-40 centrifuge, ThermoFisher) to remove the supernatant, washed once with DPBS, and pretreated with RGDS peptides or vehicle control for 30 min. Afterward, $1 \times 10^4$ LN229[TAZ(4SA)] cells were cocultured with $5 \times 10^4$ PKH26-labeled HL-60 neutrophils for 1 h in 24-well ultra-low attachment plates in the presence of RGDS or vehicle control before imaging and overlapping quantification.

## PKH26/MPO puncta quantification and immunocytochemistry (ICC)

LN229[TAZ(4SA)] cells were seeded at $2.5 \times 10^4$ cells/well on glass cover slips in 24-well flat-bottom plates and cultured overnight. Next day, $7.5 \times 10^4$ PKH26-labeled HL-60 or 32Dcl3cl3 neutrophils were added and the cells were cocultured for 30 h at 37 °C with 5% $CO_2$. For tumor-associated neutrophils, $1.0 \times 10^4$ PKH26-labeled neutrophils were added for coculture. Following coculture, cells were washed once with DPBS, and fixed in paraformaldehyde (4% in PBS) for 30 min at room temperature. Next, cells were permeabilized with PDT (0.3% sodium deoxycholate, 0.3% Triton X-100 in PBS) for 30 min on ice, blocked for 1 h with 5% BSA/PBS at room temperature, and incubated at 4 °C with primary antibodies (1:100 dilution) in 2.5% BSA/0.05% Triton X-100/PBS overnight. The following day, cells were washed three times with 0.1% Triton X-100/PBS, then incubated with a secondary antibody (1:200 dilution) in 2.5% BSA/0.05% Triton X-100/PBS for 2 h at room temperature. Cells were then washed three times with 0.1% Triton X-100/PBS, labeled with 4,6-diamidino-2-phenylindole (DAPI) (1:1000 dilution) for 5 min, washed once with PBS, then mounted on slides using ProLong Gold Antifade Mountant (P10144, Invitrogen). Samples were imaged using the Leica SP8 confocal microscope at Penn State College of Medicine's Light Microscopy Core. The optical excitations used were 405 nm, 488 nm, and 561 nm. Each sample was imaged at ~0.1 μm increments along the z-axis, and the layers were stacked using maximal intensity using ImageJ v1.5. To quantify fluorescent puncta, images were converted to 8-bit images, thresholded, and particles analyzed in ImageJ. The total puncta counts for the MPO and PKH channels were divided by the total number of cells. For experiments studying the effects of specific drugs on intercellular content transfer, GBM cells were pretreated with the respective drugs, and then cocultured in the presence of said drug at listed concentrations.

## RGDS flow cytometry assay

Zsgreen-expressing LN229[TAZ(4SA)] cells were detached from regular cell culture plates with Accutase, centrifuged for 4 min at 1000 rpm to remove the supernatant, washed once with DPBS, and pretreated with RGDS peptides or vehicle control for 30 min. Next, $1 \times 10^5$ Zs-green LN229[TAZ(4SA)] cells were cocultured with $1.5 \times 10^5$ PKH26-labeled HL-60 neutrophils for 1 h in 24-well ultra-low attachment plates and then analyzed using flow cytometry. PE emission and excitation parameters were used to analyze PKH26 signals and

FITC emission and excitation parameters were used to analyze ZsGreen signals.

## Time-lapse live-cell imaging

HL-60 or 32Dcl3cl3 neutrophils were labeled with PKH26 and cocultured with either unstained or 5-chloromethylfluorescein diacetate (CMFDA) CellTracker Green dyed GBM cells at a 1:3 ratio. The cocultures were monitored using the Incucyte S3 time-lapse imaging function. The growth rate of sh-*Vps34* and sh-control cell lines was monitored by assessing the confluence percentage of each cell line using time-lapse imaging with an initial seeding of $5 \times 10^3$ cells/well in 96-well flat-bottom plates.

## Live-cell fluorescent imaging

LN229$^{TAZ(4SA)}$ cells expressing GFP-LC3B or GFP-UVRAG were seeded at $1.5 \times 10^4$ cells/well in the center well of glass-bottom dishes (10810-054, Matsunami) and cultured overnight. The next day, $5 \times 10^4$ HL-60 neutrophils were cocultured with the GBM cells for 1 h, then imaged with IX83 Inverted microscope (Olympus) using bright field and GFP channels. Images were taken using system-optimized increments along the z-axis, and the 3D deconvolution module found in the Olympus cellSens Imaging Software was used to produce the final IF images.

## Immunoblotting

GBM cells were seeded at $3 \times 10^6$ indicated cells/well in 6-well plates and allowed to culture overnight. Cells were then lysed using SDS lysis buffer (10 mM Tris pH 7.5, 1% SDS, 50 mM NaF, 1 mM NaVO$_4$), processed in SDS-PAGE using 4–12% Bis-Tris SDS-PAGE gels (Invitrogen), and transferred to Immobilon-P membranes (Millipore). Membranes were then blocked in 5% skim milk/TBST (0.1% Tween, 10 mM Tris at pH 7.6, 100 mM NaCl) for 1 h at room temperature, rinsed once with TBST, then incubated overnight at 4 °C with primary antibodies. The following day, membrane was washed three times with TBST, and incubated in appropriate secondary HRP-conjugated antibodies at room temperature for 2 h, followed by three times of wash with TBST, then analyzed for chemiluminescence using ECL (1856136, Pierce).

## Transmission electron microscopy

For in vitro coculture samples, $5 \times 10^5$ LN229$^{TAZ(4SA)}$ cells were seeded onto sterile $60 \times 15$ mm Permanox dishes (Nalge Nunc International) and cultured overnight at 37 °C with 5% CO$_2$. The next day, 32Dcl3cl3 or HL-60 neutrophils were added onto the LN229$^{TAZ(4SA)}$ cells at a 1-to-3 ratio and allowed to coculture for 2 h at 37 °C with 5% CO$_2$. Cells were then washed two times with warm DPBS and immediately fixed with Karnovsky fixative containing 2% paraformaldehyde and 2.5% glutaraldehyde (pH 7.3) for 1 h. After fixation, the samples were submitted to the Penn State College of Medicine Transmission Electron Microscopy Core (RRID Number: SCR_021200) and processed. During processing, the samples were further fixed in 1% osmium tetroxide in 0.1 M phosphate buffer (pH 7.4) for 1 h, dehydrated in a series of graduated ethanol solutions followed by acetone, and embedded in LX-112 (Ladd Research, Williston, VT). Next, sections (60 nm) were cut from the samples and stained with uranyl acetate and lead citrate and viewed with a JEOL JEM1400 Transmission Electron Microscope (JEOL USA Inc., Peabody, MA).

## Immuno-gold electron microscopy

Immuno-gold TEM were performed using protocol previously described by (Tang et al, 2019). In brief, $3 \times 10^5$ LN229$^{TAZ(4SA)}$ cells were seeded on gridded glass-bottom dishes (P35G-1/5–14-C-GRID, MatTek) and allowed to culture overnight at 37 °C with 5% CO$_2$. The next day, $5 \times 10^5$ PKH26-labeled HL-60 neutrophils were added and allowed to coculture with the tumor cells for 1 h. Following coculture, cells were fixed for 2 h with 4% paraformaldehyde in phosphate buffer (PB), pH 7.4. Samples were then permeabilized with 0.25% saponin in PB for 30 min and then incubated for 1 h in blocking buffer (10% BSA, 10% normal goat serum, 0.1% cold water fish gelatin, 0.1% saponin in PB). Samples were then incubated overnight at 4 °C in primary MPO antibody. The next day, samples were incubated at room temperature with gold-conjugated secondary antibody (Anti-Rabbit Gold secondary antibody, NanoGold 2004) for 1 h followed by 30 min of fixation in 1% glutaraldehyde in PB. Next, samples were washed with 50 mM glycine in PBS, and treated with Original Gold Enhancement reagents (NanoGold 2113) for 3 min. Samples were then submitted to the Penn State College of Medicine Transmission Electron Microscopy Core (RRID Number: SCR_021200) and processed as described above.

## Gene expression and silencing

pBabe-neo-*TAZ* (4SA) (Lei et al, 2008) was generously provided by Dr. Kun-Liang Guan. plvx-ires-zsgreen1 was from Clontech. Lentiviral vectors encoding shRNAs targeting human *Vps34* (#94: TRCN0000037794; #96: TRCN0000037796) were used to generate *Vps34* knockdown in LN229$^{TAZ(4SA)}$ cells. Lentiviral vectors encoding shRNAs targeting human *UVRAG* (#198: TRCN0000005198; #201: TRCN0000005201) were used to generate *UVRAG* knockdown in LN229$^{TAZ(4SA)}$ cells. Lentiviral vectors encoding shRNAs targeting human *ITGA5* (#52: TRCN0000029652; #53: TRCN0000029653) were used to generate *ITGA5* knockdown in LN229$^{TAZ(4SA)}$ cells. Lentiviral vectors encoding shRNAs targeting human *ITGB1* (#45: TRCN0000029645; #48: TRCN0000029648) were used to generate *ITGB1* knockdown in LN229$^{TAZ(4SA)}$ cells. Lentiviral vectors encoding shRNAs targeting human *ITGAV* (#39: TRCN0000003239; #40: TRCN0000003240) were used to generate *ITGAV* knockdown in HL-60 cells. For qPCR to confirm knockdown status, *ITGAV* primers (Forward: ATCTGTGAGGTCGAAACAGGA; Reverse: TGGAGCATACTCAACAGTCTTTG) were used. CRISPR-Cas9 (#3 gRNA: RUBCN(h)-3-Forward CACCGTGACCCAGGATGCCTCTATG, RUBCN(h)-3-Reverse AAACCATAGAGGCATCCTGGGTCAC), (#4 gRNA: RUBCN(h)-4-Forward CACCGGATGAGCCAGTGCCTAGAGG, RUBCN(h)-4-Reverse AAACCCTCTAGGCACTGGCTCATCC) were used to generate *RUBCN* knockout in LN229$^{TAZ(4SA)}$ cells. For qPCR to confirm knockout status, *RUBCN* primers (Forward: GATTACTGGCAGTTCGTGAAAGA; Reverse: CTGCTCTGGTCGTTCTCGTG) were used. All cells expressing the shRNAs were selected via 2 µg/mL puromycin in culture media. Knockdown efficiency was examined by western blotting or qPCR once per week

following selection. All shRNAs were obtained from the PSU Genomics Sciences Core facility.

## RNA-sequencing and data processing

RNA sequencing was performed on an Illumina NovaSeq for 100 cycles using pair-end according to the manufacturer's instructions. Sequencing data were analyzed using Galaxy (https://usegalaxy.org/). Briefly, reads were trimmed and aligned to reference human genome and annotation files (GRCh38, build 38) using HISAT2. Difference in gene expression were detected by edgeR. Ingenuity Pathway Analysis (Qiagen) was then used for signaling pathway analysis based on the differential expressed genes at 2-fold and FDR < 0.05. The normalized count datasets were included in Dataset EV1.

## Immunomagnetic selection and tumor-associated neutrophil isolation

To collect tumor-associated neutrophils, terminal mice bearing LN229$^{TAZ(4SA)}$ tumors were first euthanized with carbon dioxide and cervical dislocation according to protocols approved by the institution's animal care committee. Next, tumors were resected from the mouse brain and briefly washed three times with sterile-filtered Dulbecco's phosphate-buffered saline (DPBS; 21-030-CVR, Corning). Whole tumors were then minced thoroughly and incubated with Hank's Balanced Salt Solution (HBSS; 21-022-CV, Corning) with 0.1 mg/mL type IV collagenase (C5138, Sigma) and 0.1 mg/mL hyaluronidase (H6254, Sigma) at 37 °C for 15 min. To stop the digestion, equal volumes of HBSS containing 5% FBS were added. The cell mixture was centrifuged for 5 min at 1000 rpm (110 × $g$; ST-40 centrifuge, ThermoFisher). Cells were washed twice using HBSS with 5% FBS mixture, with the same centrifuge settings, to remove any remaining digestive enzymes. The cell mixture was then filtered through sterile 40 μm nylon mesh cell strainer (22-363-547, Thermo-Fisher). Cells were resuspended in red blood cell lysis buffer (11814389001, Sigma) to lyse red blood cells and then washed three times in RPMI with 5% FBS. Cells were loaded with trypan blue, counted on an automated cell counter, and resuspended in DPBS with 5% FBS for use with immunomagnetic selection. Mouse neutrophil isolation kit (130-097-658, Miltenyi Biotec) was used to isolate tumor-associated neutrophils from the cell mixture according to the manufacturer's instructions. In brief, singly suspended live cells were first incubated in purified rat anti-mouse CD16/32 Fc receptor block (553141, BD Biosciences) and then labeled with a neutrophil biotin-antibody cocktail containing biotin-conjugated monoclonal antibodies against antigens not expressed on neutrophils. The labeled cells were selected out of the population using anti-biotin microbeads and MS columns (130-042-201, Miltenyi Biotec). The flow-through containing tumor-associated neutrophils had purities higher than 90% as evaluated by flow cytometry.

## Statistics and reproducibility

For statistical analyses, most of the experiments were repeated three or more times as indicated in each figure legend. No data were excluded from the analyses. In the in vitro experiments, all cells in each experiment were from the same pool of parental cells. All mice in different groups in each experiment were from the same cohort. Investigators were not blinded during data collection and group allocation. Statistical significance was tested as indicated in each figure legend. All values shown are mean values, and all error bars represent standard errors of the mean (s.e.m.). GraphPad Prism 10 was used to plot and perform the statistical analysis. For in vivo studies, each data point indicates one animal. For in vitro studies, each data point indicates the average of technical replicates from independent experiments. The number of independent repeats ($n$) is indicated in figure legends. For imaging studies requiring the quantification of cells for overlap and engulfment stages, around 300 cells were counted for each group for each independent replicate. For immunofluorescent imaging studies for PKH26 and MPO quantification, at least 2 representative fields of views were quantified per group, per independent replicate.

## Data availability

The RNA-seq datasets from this study have been deposited to the Gene Expression Omnibus (https://www.ncbi.nlm.nih.gov/geo/query/acc.cgi?acc=GSE263766) with the accession # GSE263766.

The source data of this paper are collected in the following database record: biostudies:S-SCDT-10_1038-S44318-024-00130-4.

## Peer review information

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

## Acknowledgements

We would like to thank members of the Li laboratory for helpful discussions; the Transmission Electron Microscopy Core (RRID: SCR_021200) and the Microscopy Imaging Core (RRID: SCR_022526) (Leica SP8 Confocal: 1S10OD010756-01A1 CB); Dr. Wesley Raup-Konsavage from the Drug Discovery, Development and Delivery core; Mr. Nate Sheaffer, Ms. Jade Vogel, and Mr. Joseph Bednarczyk from the Flow Cytometry Core (RRID: SCR_021134); Dr. Sirisha Pochareddy from the Genomics Sciences Core (RRID: SCR_021123); Ms. Gretchen Snavely and Ms. Erin Mattern from the Comparative Medicine Histopathology Core; and Dr. Nataliya Smith, Ms. Kristin Shuler and Mr. John Graybeal from the Department of Neurosurgery's Neuroscience Research Institute Biorepository for assistance with sample handling and IRB submissions. We acknowledge support from the National Institutes of Neurological Disorders and Stroke (R01 NS109147 and NS119547 to WL), Penn State College of Medicine Medical Scientist Training Program (5T32GM118294 to PY through PSU), and the Four Diamonds (to PSU).

## Author contributions

**Tong Lu**: Conceptualization; Data curation; Formal analysis; Investigation; Methodology; Writing—original draft; Writing—review and editing. **Patricia P Ye**: Conceptualization; Data curation; Formal analysis; Investigation; Methodology; Writing—original draft; Writing—review and editing. **Stephen Y Chih**: Investigation; Methodology; Writing—review and editing. **Miaolu Tang**: Investigation; Methodology; Writing—review and editing. **Han Chen**: Investigation; Methodology; Writing—review and editing. **Dawit G Aregawi**: Resources; Writing—original draft; Writing—review and editing. **Michael J Glantz**: Resources; Writing—original draft; Writing—review and editing. **Brad E Zacharia**: Resources; Writing—original draft; Writing—review and editing. **Hong-Gang Wang**: Resources; Writing—original draft; Writing—review and editing. **Wei Li**: Conceptualization; Resources; Data curation; Formal analysis; Supervision; Funding acquisition; Validation; Investigation; Visualization; Methodology; Writing—original draft; Project administration; Writing—review and editing.

Source data underlying figure panels in this paper may have individual authorship assigned. Where available, figure panel/source data authorship is listed in the following database record: biostudies:S-SCDT-10_1038-S44318-024-00130-4.

## Disclosure and competing interests statement

The authors declare no competing interests.

# Expanded View Figures

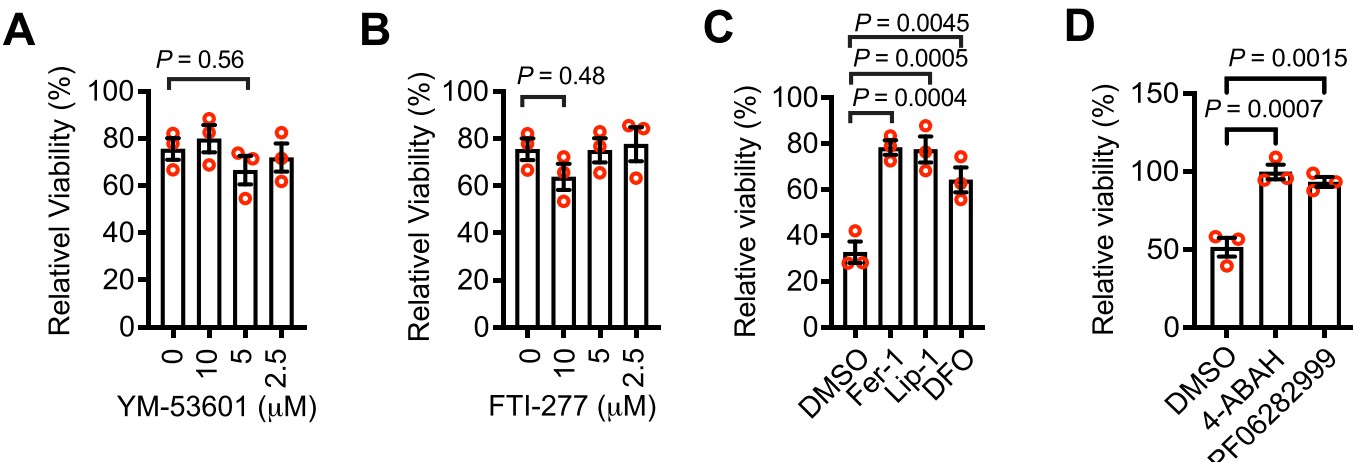

**Figure EV1.  Viability of LN229$^{TAZ(4SA)}$ cells cocultured with dHL-60 cells in the presence of various inhibitors.**

(A, B) Luciferase assay results for LN229$^{TAZ(4SA)}$ cells cocultured with dHL-60 cells with YM-53691 ($n = 3$ independent experiments), and FTI-277 ($n = 3$ independent experiments). Luminescence readouts of cocultured wells were normalized to their respective LN229$^{TAZ(4SA)}$ cells monocultured wells, both with drugs. One-way ANOVA. (C) Luciferase assay results for LN229$^{TAZ(4SA)}$ cells cocultured with dHL-60 cells with ferrostatin-1 (2 µM), liproxstatin-1 (0.2 µM), and DFO (0.2 mM). Luminescence readouts of cocultured wells were normalized to their respective LN229$^{TAZ(4SA)}$ cells monocultured wells, both with drugs ($n = 3$ independent experiments). One-way ANOVA. (D) Luciferase assay results for LN229$^{TAZ(4SA)}$ cells cocultured with dHL-60 cells with 4-ABAH (2 µM) or PF06282999 (2 µM). Luminescence readouts of cocultured wells were normalized to their respective LN229$^{TAZ(4SA)}$ cells monocultured wells, both with drugs ($n = 3$ independent experiments). One-way ANOVA. Error bars, s.e.m.

 

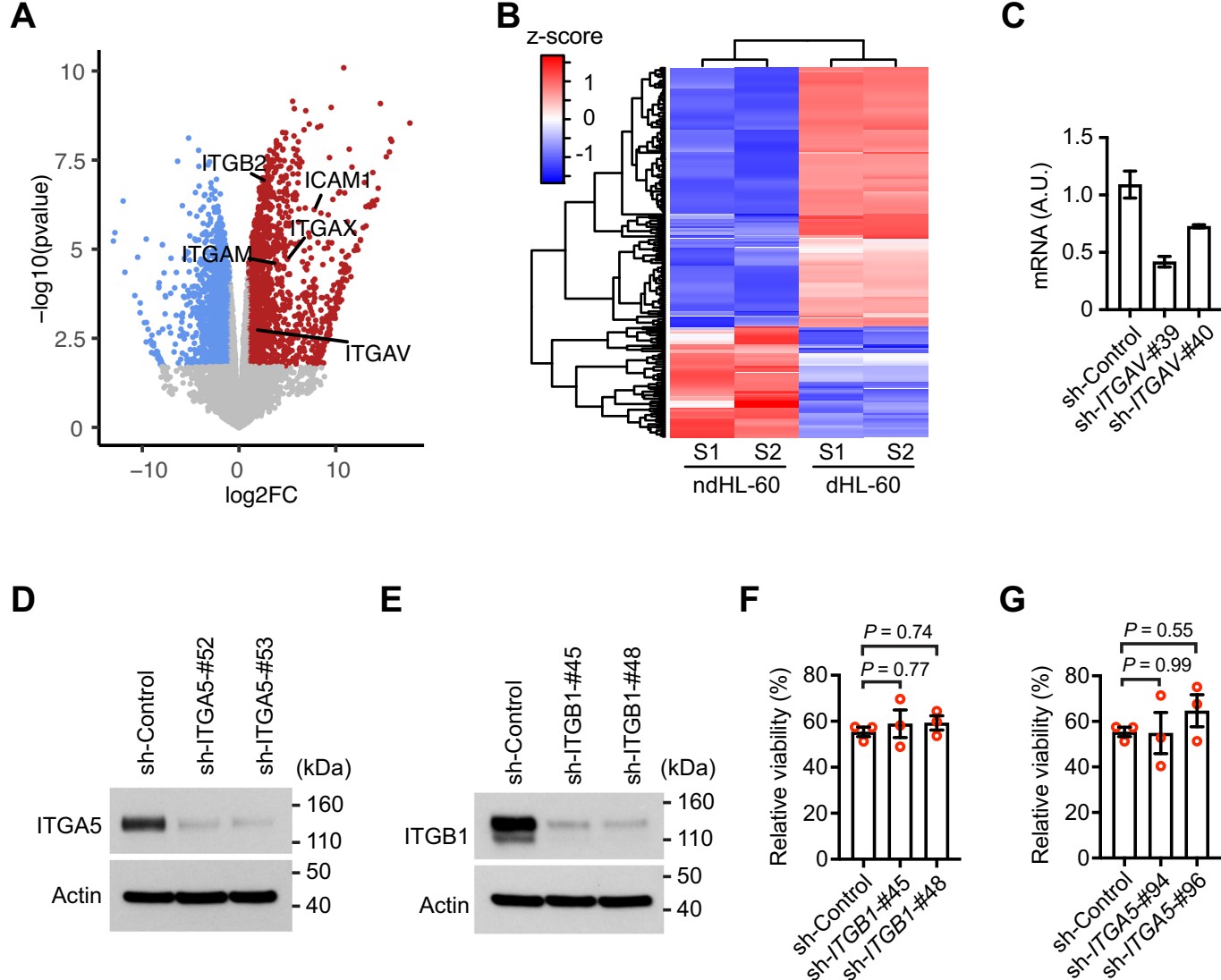

**Figure EV2.** **Integrin expression and function in the coculture of LN229[TAZ(4SA)] cells and dHL-60 cells.**

(A) Volcano plot of differentially expressed genes in dHL-60 cells compared to ndHL-60 cells. Upregulated integrin family genes are labeled. The Benjamini-Hochberg False Discovery Rate (FDR) test. (B) Heatmap plot of differentially expressed genes related to the neutrophil degranulation pathway. Two replicate samples (S1, S2) were analyzed in each condition. (C) dHL-60 cells transduced by indicated shRNAs were subjected to qRT-PCR ($n = 2$ technical replicates). (D, E) LN229[TAZ(4SA)] cells transduced by indicated shRNAs were subjected to Western blotting. (F, G) Cancer cell viability determined using luciferase readouts for LN229[TAZ(4SA)] cells transduced with indicated shRNAs cocultured with dHL-60 cells. Luminescence readouts of cocultured wells were normalized to their respective monocultured wells for each LN229[TAZ(4SA)] cell line ($n = 3$ independent experiments). One-way ANOVA. Error bars, s.e.m.

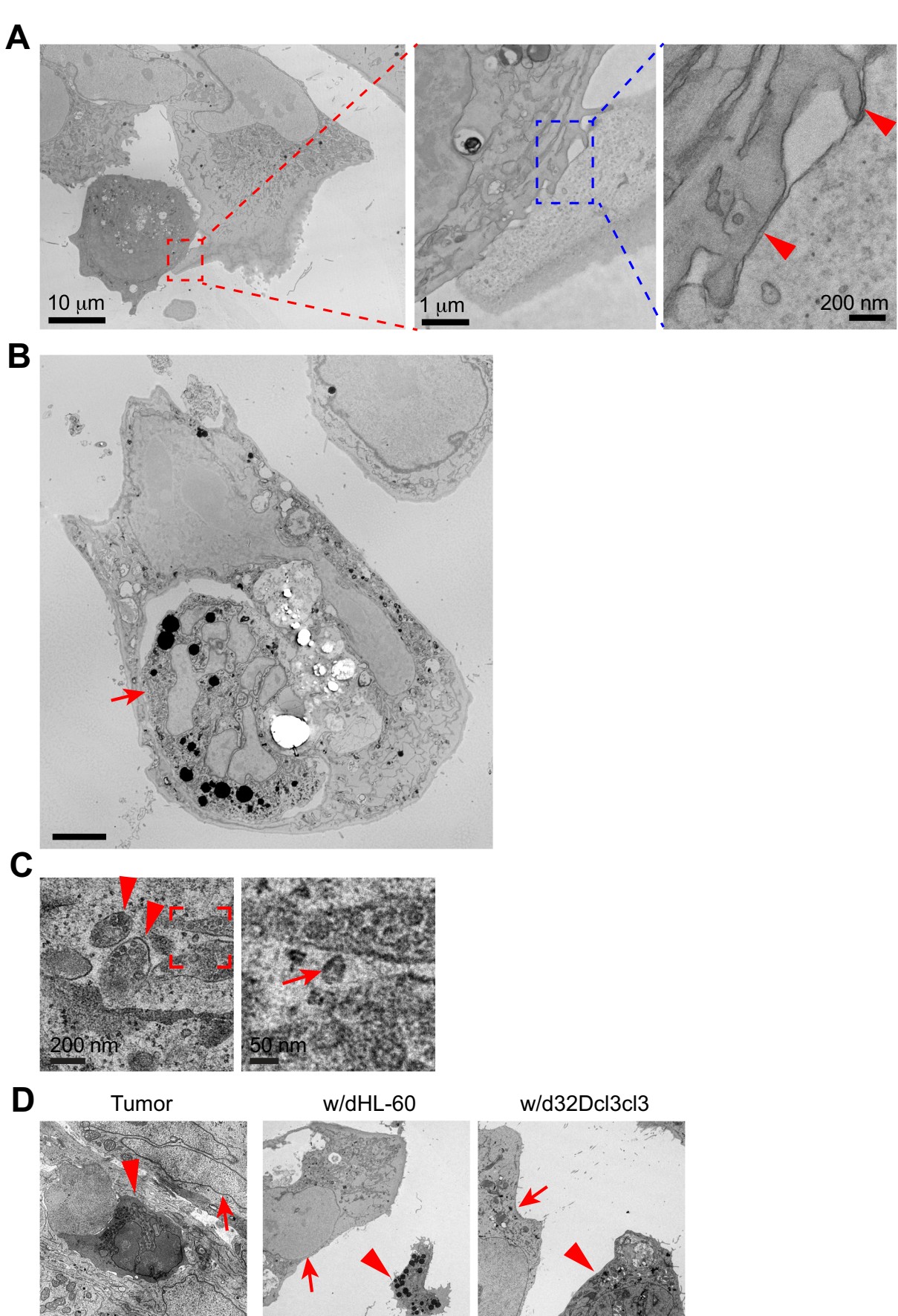

◀

**Figure EV3.  TEM analysis of the coculture of LN229TAZ(4SA) cells and neutrophils.**

(A) Low- and high-magnification transmission electron microscopy (TEM) images of LN229$^{TAZ(4SA)}$ cells cocultured with dHL-60 neutrophils. The outlined areas are sequentially enlarged and shown as indicated. Arrowheads point adhesion between a tumor cell and a neutrophil. (B) TEM images LN229$^{TAZ(4SA)}$ cells cocultured with dHL-60 neutrophils. The arrow points an engulfed neutrophil. Scale bar, 5 µm. (C) TEM images of LN229$^{TAZ(4SA)}$ tumor cells cultured with d32Dcl3 neutrophils. The outlined area is shown on the right. Arrowheads point to larger granules. The arrow points to smaller granules. (D) TEM images of LN229$^{TAZ(4SA)}$ tumor sections (left), LN229$^{TAZ(4SA)}$ tumor cells cultured with dHL-60 cells (middle), or d32Dcl3cl3 cells (left). Tumor cells (arrows). Neutrophils (arrowheads). Scale bars, 5 µm.

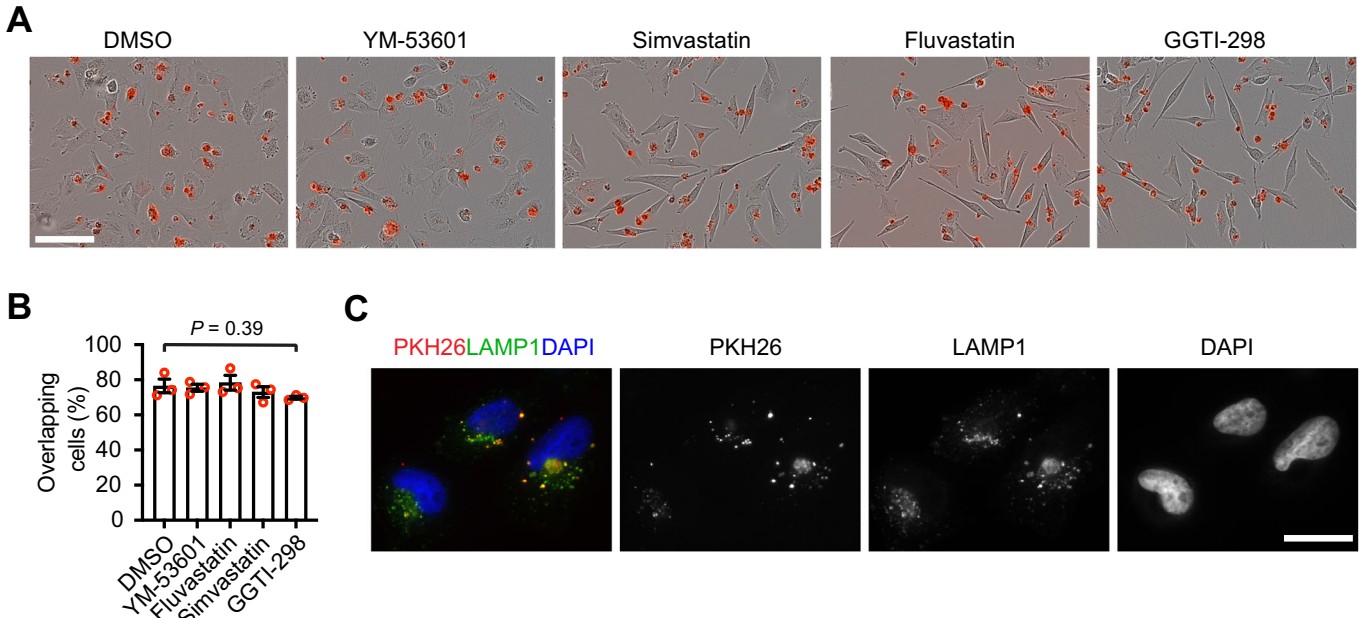

**Figure EV4. Internalization of neutrophils by tumor cells.**

(A) Representative images showing LN229$^{TAZ(4SA)}$ cells cocultured with PKH26-labeled dHL-60 cells with DMSO, simvastatin (2.5 μM), fluvastatin (2.5 μM), GGTI-298 (10 μM), or YM-53601 (10 μM). Scale bar, 100 μm. (B) Quantification of LN229$^{TAZ(4SA)}$ cells with overlapping PKH26-labeled dHL-60 cells in the conditions described in (A) ($n = 3$ independent experiments). One-way ANOVA. (C) Representative immunofluorescent images showing LAMP1 staining, PKH26 signal, and DAPI staining for LN229$^{TAZ(4SA)}$ cells cocultured with PKH26-labeled d32Dcl3cl3 cells ($n = 3$). Scale bar, 20 μm. Error bars, s.e.m.

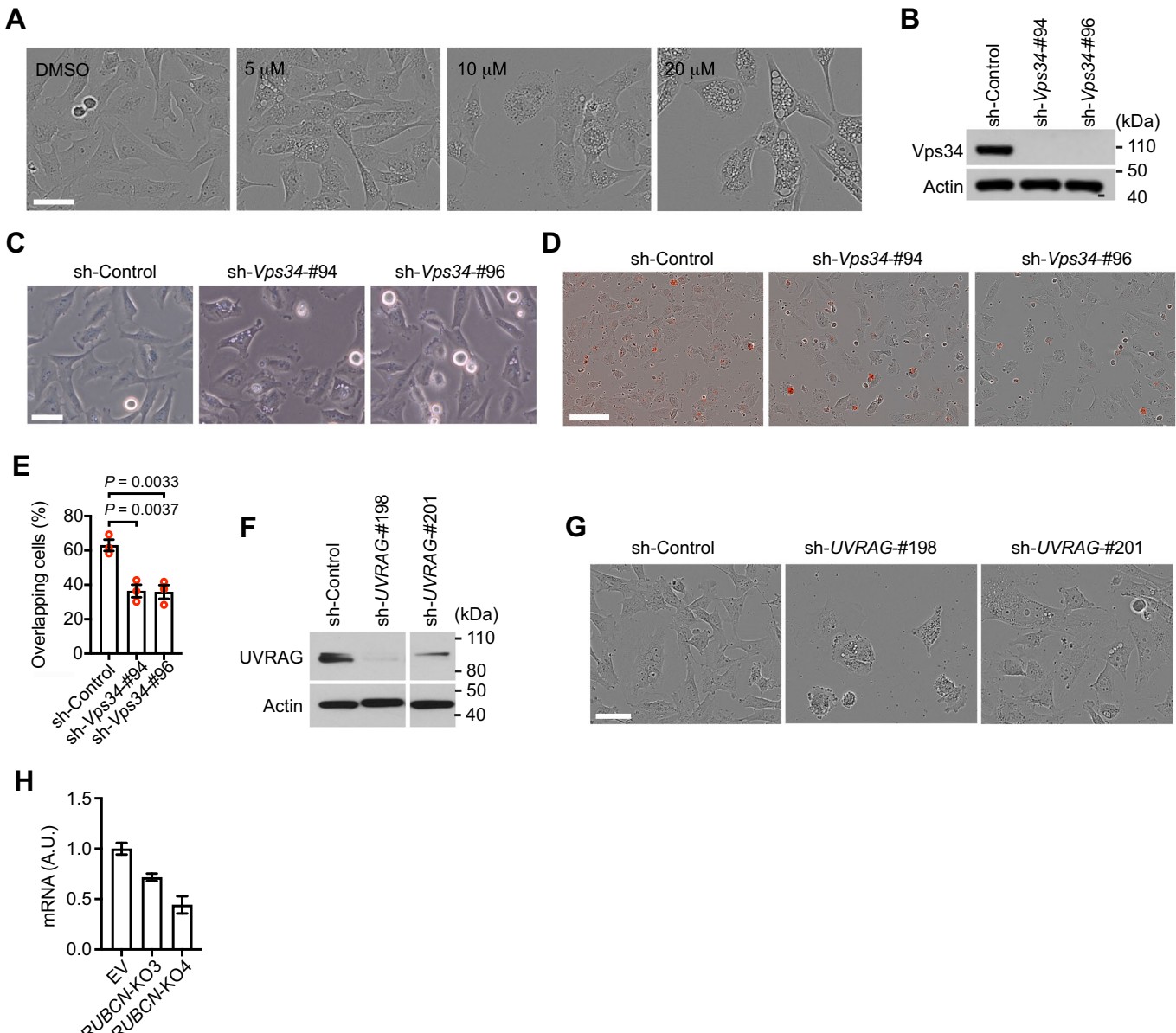

**Figure EV5. Characterization of LN229^TAZ(4SA) cells upon knocking down various LAP genes.**

(A) Representative images showing the morphology of LN229^TAZ(4SA) cells after 12 h treatment with Vps34-IN1 at listed concentrations. Scale bar, 50 μm. (B) LN229^TAZ(4SA) cells transduced by indicated shRNAs were subjected to Western blotting. (C) Representative images showing the morphology of LN229^TAZ(4SA) cells transduced by indicated shRNAs. Scale bar, 50 μm. (D) Representative images showing the morphology of LN229^TAZ(4SA) cells transduced by indicated shRNAs cocultured with PKH26-labeled dHL-60 cells (*n* = 3). Scale bar, 100 μm. (E) Quantification of LN229^TAZ(4SA) cells with overlapping PKH26-labeled dHL-60 cells for groups indicated in (D) (*n* = 3 independent experiments). One-way ANOVA. (F) LN229^TAZ(4SA) cells transduced by indicated shRNAs were subjected to Western blotting. (G) Representative images showing the morphology of LN229^TAZ(4SA) cells transduced by indicated shRNAs. Scale bar, 50 μm. (H) LN229^TAZ(4SA) cells transduced by indicated gRNAs were subjected to qRT-PCR (*n* = 2 technical replicates). Error bars, s.e.m.

