## [Peer Review File · The EMBO Journal]

LC3-associated phagocytosis of neutrophils triggers tumor ferroptotic cell death in glioblastoma

Wei Li, Tong Lu, Patricia Yee, Stephen Chih, Miaolu Tang, Han Chen, Dawit Aregawi, Michael Glantz, Brad Zacharia, and Hong-Gang Wang

Corresponding author: Wei Li (weili@pennstatehealth.psu.edu)

Review Timeline:

Submission Date:	22nd Oct 23
Editorial Decision:	19th Dec 23
Revision Received:	18th Mar 24
Editorial Decision:	6th Apr 24
Revision Received:	24th Apr 24
Accepted:	30th Apr 24

Editor: Daniel Klimmeck

Transaction Report:

Dear Dr Li,

Thank you again for submitting your manuscript EMBOJ-2023-115956 for consideration by the EMBO Journal. Please accept my sincere apologies for getting back to you with unusual protraction due to delayed referee input, as well as detailed discussion in the editorial team. As indicated, your manuscript has been seen by three referees with expertise in glioblastoma biology, cancer immunology and cell death and we have received reports from all of them, which are shown below.

In light of the referees' overall encouraging recommendations, I would like to invite you to submit a revised version of the manuscript, addressing the comments of all three reviewers. I should add that it is EMBO Journal policy to allow only a single round of revision, and acceptance of your manuscript will therefore depend on the completeness of your responses in this revised version.

Given the numerous issues raised, I would appreciate if you could contact me during the next weeks for exchange e.g. a video call to discuss your perspective on the comments and potential plan for revisions.

Please feel free to contact me if you have any questions or need further input on the referee comments.

When submitting your revised manuscript, please carefully review the instructions below.

Please feel free to approach me any time should you have additional questions related to this.

Thank you for the opportunity to consider your work for publication.

I look forward to your revision.

Kind regards,

Daniel Klimmeck

Daniel Klimmeck, PhD
Senior Editor
The EMBO Journal

Instruction for the preparation of your revised manuscript:

2) individual production quality figure files as .eps, .tif, .jpg (one file per figure).

3) a .docx formatted letter INCLUDING the reviewers' reports and your detailed point-by-point response to their comments. As part of the EMBO Press transparent editorial process, the point-by-point response is part of the Review Process File (RPF), which will be published alongside your paper.

4) a complete author checklist, which you can download from our author guidelines ([https://wol-prod-cdn.literatumonline.com/pb-assets/embo-site/Author Checklist%20-%20EMBO%20J-1561436015657.xlsx](https://wol-prod-cdn.literatumonline.com/pb-assets/embo-site/Author%20Checklist%20-%20EMBO%20J-1561436015657.xlsx)). Please insert information in the checklist that is also reflected in the manuscript. The completed author checklist will also be part of the RPF.

6) It is mandatory to include a 'Data Availability' section after the Materials and Methods. Before submitting your revision, primary datasets produced in this study need to be deposited in an appropriate public database, and the accession numbers and database listed under 'Data Availability'. Please remember to provide a reviewer password if the datasets are not yet public (see <https://www.embopress.org/page/journal/14602075/authorguide#datadeposition>).

7) Our journal encourages inclusion of *data citations in the reference list* to directly cite datasets that were re-used and obtained from public databases. Data citations in the article text are distinct from normal bibliographical citations and should directly link to the database records from which the data can be accessed. In the main text, data citations are formatted as follows: "Data ref: Smith et al, 2001" or "Data ref: NCBI Sequence Read Archive PRJNA342805, 2017". In the Reference list, data citations must be labelled with "[DATASET]". A data reference must provide the database name, accession number/identifiers and a resolvable link to the landing page from which the data can be accessed at the end of the reference. Further instructions are available at .

8) At EMBO Press we ask authors to provide source data for the main and EV figures. Our source data coordinator will contact you to discuss which figure panels we would need source data for and will also provide you with helpful tips on how to upload and organize the files.

Numerical data can be provided as individual .xls or .csv files (including a tab describing the data). For 'blots' or microscopy, uncropped images should be submitted (using a zip archive or a single pdf per main figure if multiple images need to be supplied for one panel). Additional information on source data and instruction on how to label the files are available at .

9) We replaced Supplementary Information with Expanded View (EV) Figures and Tables that are collapsible/expandable online (see examples in <https://www.embopress.org/doi/10.15252/emj.201695874>). A maximum of 5 EV Figures can be typeset. EV Figures should be cited as 'Figure EV1, Figure EV2' etc. in the text and their respective legends should be included in the main text after the legends of regular figures.

11) For data quantification: please specify the name of the statistical test used to generate error bars and P values, the number (n) of independent experiments (specify technical or biological replicates) underlying each data point and the test used to calculate p-values in each figure legend. The figure legends should contain a basic description of n, P and the test applied. Graphs must include a description of the bars and the error bars (s.d., s.e.m.).

The revision must be submitted online within 90 days; please click on the link below to submit the revision online before 18th Mar 2024.

Referee #1:

Extending the group's previous studies, Lu et al report that neutrophils induce ferroptosis in glioblastoma (GBM) cells following their LAP-mediated engulfment and transfer of granule content, predominantly myeloperoxidase. They also show that pharmacological inhibition of MPO, ferroptosis with liproxstatin-1 or LAP through Vps34 depletion reduce necrosis in GBM and prolong survival of tumor bearing mice. Furthermore, various statins to varying degrees can also inhibit neutrophil-mediated GBM killing, a critical component of GBM progression. These findings identify a novel mechanism by which neutrophils may modulate tumor progression. There are several issues that may require further attention.

Specific Comments.

1. Inhibition of LAP inhibited necrosis and prolonged survival, but had little effect on tumor growth. These observations would imply that tumor growth might occur independent of necrosis, which contradicts the authors' central hypothesis. How could these be reconciled?
2. Liproxstatin-1 blocked necrosis in GBM and attenuated tumor growth, but had no survival benefit. Thus, necrosis in GBM appears to be a poor predictor of outcome. These warrant further consideration.
3. A potential limitation of the study is the use of cancer cell lines to generate neutrophil-like cells. Thus, repeating the key experiments with human or mouse primary neutrophils would lend additional support to the authors' concept.
4. The authors' screen identified statins as potential inhibitor of neutrophil-mediated ferroptosis. While the findings are described in detail, statins (and differences in their actions) are not discussed at all.
5. The text often lacks clarity in regard of the use of naïve or differentiated HL-60 and 32Dcl3cl3 cells. Statistically not-significant changes should not be referred to as "trends".
6. The terms necrosis and pyroptosis are sometimes used interchangeably, this should be clarified throughout the text.

Minor Points.

1. Myeloperoxidase is expressed almost exclusively in neutrophils (and at much lower levels in monocytes). Hence, reference to "neutrophilic MPO" is superfluous.
2. The authors should carefully proofread the manuscript for typos (e.g. "labeled" is misspelled as "labled" throughout the text) and fragmented sentences (e.g. figure legends).
3. What does "ordinary one-way ANOVA" refer to?
4. Fig.7i. A brief caption would help the reader to better appreciate the working model.

Referee #2:

This paper is a continuation of "Neutrophil-induced ferroptosis promotes tumor necrosis in glioblastoma progression" by the same group published in nature communications in 2020. The initial study demonstrated that co-culturing neutrophils with glioblastoma cells (GBM) induces tumor cell ferroptosis, dependent on lipid peroxidases accumulation and MPO transfer. This current study aims to elucidate how MPO is transferred into tumors. The authors discovered through small molecule screening that neutrophils in direct contact with GBM cells are completely engulfed and fractionated by the latter via LC3-associated phagocytosis (LAP), dependent on the vps34-uvrag-pi3k axis. The notion of neutrophils committing 'suicide' to be actively engulfed by tumor cells is interesting. However, many conclusions are not well supported. The paper focuses on differentiated neutrophil-like cells, HL-60 and 32dlc3, highlighting the well-known increase in integrins after differentiation as the mechanism of cell contact, but the results are not entirely convincing. Additionally, despite the previous paper's emphasis on MPO-induced lipid peroxidation buildup, this study lacks solid results on tumor cell ferroptosis. The only experiment with liproxstatin showed a mild, statistically insignificant decrease in neutrophil-mediated tumor cell death, which does not exclude the possibility of ROS-induced cell death. Therefore, it's unclear whether this phagocytosis of neutrophil-induced cell death is a parallel and independent process from MPO-induced cell death. This paper's value will also lie in determining which neutrophil type or stage facilitates this process and testing this hypothesis in an in vivo model.

Major comments:

The major flaw is the lack of direct evidence demonstrating that LAP is necessary for the transfer of MPO-containing granules and the induction of cancer cell death by neutrophils.

Figure 2: The necessity of cell contact (using HL-60 and 32DLC3) doesn't exclude vesicle transmission. Demonstrate ferroptosis specificity with Ferrostatin rescue (Fig 2d/e).

Figure 3: HL-60 cells are known to upregulate integrin and adhesion molecules upon differentiation. Using a total integrin blocker might affect multiple adhesion pathways, not just the molecule responsible for this contact. An integrin knockout HL-60 cell line is required for precise elucidation. If non-specific cell contact is crucial, and not a particular integrin, then RGDS-treated neutrophils should still affect tumor killing. However, figures 3g and 3h show only a 20% decrease in tumor cell uptake of neutrophil matter but a dramatic increase in tumor cell viability (fig 3j), suggesting more than just contact is involved.

Figure 4: Why would neutrophils 'commit suicide' in this process if they initiate it to be consumed? Could it involve aged neutrophils or a subtype? There needs to be proof that the dense EM granules are of neutrophil origin and contain MPO.

The inhibition of MPO or depletion of Vps34 in an orthotopic xenograft GBM mouse model reduced necrosis formation and increased the survival of tumor-bearing mice. However, these effects may not be mediated by internalized neutrophils and are unrelated to LAP.

It's hypothesized that myeloperoxidase (MPO)-containing granules in tumor cells originate from internalized neutrophils. However, no data supports this. The proportion of MPO-containing granules in tumor cells derived from this route is unclear. It is also not clear how MPO-containing granules are produced by the internalized neutrophils.

The study presents no data to show that HL-60 cell-induced tumor cell death is mediated by MPO-containing granules.

HL-60 and 32dlc3-derived "neutrophils" differ significantly from primary neutrophils.

Referee #3:

This work represents a significant addition to the emerging studies in the field, which suggest that tumor-associated neutrophils contribute to glioblastoma progression by increasing intra-tumoral necrosis.

In the present study, the authors performed an unbiased small-molecule screen and found that statins reduce a pro-necrotic interaction between neutrophils and LN229 glioblastoma cells. The most effective fluvastatin and atorvastatin strongly inhibited the neutrophil-mediated tumor cell killing. This effect involved activity of the statin-sensitive mevalonate pathway, required the LC3-dependent phagocytosis of neutrophils by tumor cells, and was possibly dependent on ferroptosis (see comments below). Inhibitor of neutrophil myeloperoxidase and RNAi-driven depletion of Vps34-UVRAG signaling molecules in cancer cells both alleviated the neutrophil-driven glioblastoma necrosis. The same treatments extended animal survival in an orthotopic xenograft tumor model.

Overall, the manuscript makes a strong impression. This study adds to our understanding of glioblastoma biology and carries significant therapeutic relevance. It is expected that this work will be of significant interest to the field and the general readership of the EMBO Journal. Nevertheless, there are a few questions that require additional attention.

Major and moderate comments:

[1] Major issue: Perhaps I am missing something, but all critical in vitro and in vivo findings of this work were collected using LN229 glioblastoma cell line. It is not clear in which experiments (if any) the Authors used the patient derived primary GBM tumor cells which are described in Materials and Methods.

[2] Moderate concern: The Authors propose ferroptosis as the main mechanism for the neutrophil-induced glioblastoma cell death. I appreciate the previous publication from this group (Yee et al., 2020), which tested the mechanistic contributions of ferroptosis. Yet, in the present study, the evidence for the involvement of ferroptosis is rather rudimentary. I see only the effect of liproxstatin-1 and then it is not entirely conclusive. For this reason, I would advise against mentioning ferroptosis in abstract and the first paragraph of discussion (stay with necrosis). I also suggest re-emphasizing the previous work and the role of ferroptosis as discussion progresses. Perhaps an additional reference to the prior work is also warranted for schematic diagram presented in Fig. 7i.

Technical comment:

[3] There is at least one instance of referring to wrong figure. On p. 7, while describing EM morphology of internalized granules, the authors refer to Fig. 5g, while the information-in-question is presented in Fig. 4g. Please double-check your text for additional typos of this nature.

Response to reviewers' comments

REVIEWER COMMENTS

Referee #1:

Extending the group's previous studies, Lu et al report that neutrophils induce ferroptosis in glioblastoma (GBM) cells following their LAP-mediated engulfment and transfer of granule content, predominantly myeloperoxidase. They also show that pharmacological inhibition of MPO, ferroptosis with liproxstatin-1 or LAP through Vps34 depletion reduce necrosis in GBM and prolong survival of tumor bearing mice. Furthermore, various statins to varying degrees can also inhibit neutrophil-mediated GBM killing, a critical component of GBM progression. These findings identify a novel mechanism by which neutrophils may modulate tumor progression. There are several issues that may require further attention.

Specific Comments.

1. Inhibition of LAP inhibited necrosis and prolonged survival, but had little effect on tumor growth. These observations would imply that tumor growth might occur independent of necrosis, which contradicts the authors' central hypothesis. How could these be reconciled?

There may be a confusion when comparing the results of tumor growth with survival. Tumor growth in Figure 7E was measured in day 13 and 19 after tumor implantation (ATI). At this stage, necrosis is still much smaller, especially up to day 16 ATI than that after day 20 ATI (Yee *et al*, 2022). Therefore, the effect of LAP inhibition on tumor necrosis and growth could be undetectable. In this model, tumor necrosis progressively becomes prominent after day 20 ATI. Therefore, we predict that the effect of necrosis inhibition on tumor progression would be more prominent at this later stage (day 20 ATI and after). However, because mice from the control group started to reach the terminal stage due to tumor burden around day 20 ATI, we could not further measure the tumor growth in this group. Therefore, results in Figure 7E could not reflect tumor growth at the stage when necrosis become more prominent, whereas survival results covered the full tumor development course and could indicate the benefit of LAP inhibition. We have added a clarification in revised manuscript Page 12.

2. Liproxstatin-1 blocked necrosis in GBM and attenuated tumor growth, but had no survival benefit. Thus, necrosis in GBM appears to be a poor predictor of outcome. These warrant further consideration.

We agree with the reviewer that necrosis blockade and ineffective survival improvement by liproxstatin-1 appear to be inconsistent with our hypothesis. Considering these results and our previously published results, we would suggest to interpretate the results cautiously. Our

previous results showed that ectopic expression of GPX4 (a ferroptosis suppressor) or depletion of ACSL4 (an essential ferroptosis component) can reduce necrosis by 47% or 58-68%, respectively. These genetic manipulations can effectively prolong tumor-bearing mice survival (Figure 6b-e in (Yee *et al*, 2020)). Comparing to these genetic manipulations, the effect of liproxstatin-1 on necrosis blockade is less (reducing necrosis by 29%, yet insignificant). The differential efficacy on necrosis blockade in between these genetic and pharmacologic approaches may explain the differential effects on survival. Considering these, the liproxstatin-1 results included in the current manuscript may not lead to a conclusion that necrosis in GBM is a poor predictor of outcome. We have added a note in revised manuscript Page 11-12.

3. A potential limitation of the study is the use of cancer cell lines to generate neutrophil-like cells. Thus, repeating the key experiments with human or mouse primary neutrophils would lend additional support to the authors` concept.

We have isolated primary neutrophils from LN229^{TAZ(4SA)} cell-derived tumors and used these cells to examine and confirm several key conclusions. The results using these cells have been included in revised Figures 1F, 1G, 3K, 6E, 6G, 6H-L, and 6P.

4. The authors' screen identified statins as potential inhibitor of neutrophil-mediated ferroptosis. While the findings are described in detail, statins (and differences in their actions) are not discussed at all.

We have added statins-related discussion in revised manuscript Page 13-14.

5. The text often lacks clarity in regard of the use of naïve or differentiated HL-60 and 32Dcl3cl3 cells. Statistically not-significant changes should not be referred to as "trends".

We have clarified the terms by referring HL-60 cells at non-differentiated or naive state to ndHL-60, while using dHL-60 and d32Dcl3cl3 to refer differentiated HL-60 and 32Dcl3cl3 cells, respectively.

We have modified the description of statistically not-significant changes by using “not significantly.”

6. The terms necrosis and pyroptosis are sometimes used interchangeably, this should be clarified throughout the text.

We have clarified the term usage.

Minor Points.

1. Myeloperoxidase is expressed almost exclusively in neutrophils (and at much lower levels in monocytes). Hence, reference to "neutrophilic MPO" is superfluous.

We have deleted neutrophilic and use MPO only.

2. The authors should carefully proofread the manuscript for typos (e.g. "labeled" is misspelled as "labled" throughout the text) and fragmented sentences (e.g. figure legends).

We thank the reviewer for catching these mistakes and have corrected them.

3. What does "ordinary one-way ANOVA" refer to?

We have changed to "one-way ANOVA".

4. Fig. 7i. A brief caption would help the reader to better appreciate the working model.

We have added a caption for Figure 7I in the revised figure legend.

Referee #2:

This paper is a continuation of "Neutrophil-induced ferroptosis promotes tumor necrosis in glioblastoma progression" by the same group published in nature communications in 2020. The initial study demonstrated that co-culturing neutrophils with glioblastoma cells (GBM) induces tumor cell ferroptosis, dependent on lipid peroxidases accumulation and MPO transfer. This current study aims to elucidate how MPO is transferred into tumors. The authors discovered through small molecule screening that neutrophils in direct contact with GBM cells are completely engulfed and fractionated by the latter via LC3-associated phagocytosis (LAP), dependent on the vps34-uvrag-pi3k axis. The notion of neutrophils committing 'suicide' to be actively engulfed by tumor cells is interesting. However, many conclusions are not well supported. The paper focuses on differentiated neutrophil-like cells, HL-60 and 32dlc3, highlighting the well-known increase in integrins after differentiation as the mechanism of cell contact, but the results are not entirely convincing. Additionally, despite the previous paper's emphasis on MPO-induced lipid peroxidation buildup, this study lacks solid results on tumor cell ferroptosis. The only experiment with liproxstatin showed a mild, statistically insignificant decrease in neutrophil-mediated tumor cell death, which does not exclude the possibility of ROS-induced cell death. Therefore, it's unclear whether this phagocytosis of neutrophil-induced cell death is a parallel and independent process from MPO-induced cell death. This paper's value will also lie in determining which neutrophil type or stage facilitates this process and testing this hypothesis in an in vivo model.

Major comments:

The major flaw is the lack of direct evidence demonstrating that LAP is necessary for the transfer of MPO-containing granules and the induction of cancer cell death by neutrophils.

To examine the requirement of LAP for the transfer of MPO-containing granules, we have assessed MPO⁺ granules in LN229^{TAZ(4SA)} tumor cells cocultured with tumor-associated neutrophils upon depleting Vps34 through shRNAs or inhibiting Vps34 by its inhibitor Vps34-IN1. As shown in the revised Figures 6H-L, MPO⁺ granules in tumor cells are much fewer in Vps34-depleted or -inhibited cells than control cells. These results supported that LAP is essential for the transfer of MPO-containing granules. We also examined MPO⁺ granules when adding statins and the MVA pathway inhibitors into the coculture of LN229^{TAZ(4SA)} tumor cells and dHL-60 cells. As shown in the revised Figures 1J-L, MPO⁺ granules in tumor cells are also fewer in statins- or GGTI-298-treated cells. These results also supported that MPO-containing granules and PKH⁺ granules are transferred to tumor cells in parallel.

To further examine the requirement of LAP for the induction of cancer cell death by neutrophils, we have examined the dependency on RUBCN, a gene uniquely required for LAP. Depletion of RUBCN increased survival of LN229^{TAZ(4SA)} tumor cells when they were cocultured with dHL-60 cells or tumor-associated neutrophils (revised Figure 6O and 6P). In addition, we have also examined the role of Vps34 in cancer cell death induced by tumor-associated neutrophils. As shown in revised Figure 6E and 6G, depletion or inhibition of Vps34 can increase the survival of LN229^{TAZ(4SA)} tumor cells when they are cocultured with tumor-associated neutrophils. Overall, we have demonstrated that three LAP genes, including Vps34, UVRAG and RUBCN, are essential for tumor cell death induced by dHL-60 or tumor-associated neutrophils.

Figure 2: The necessity of cell contact (using HL-60 and 32DLC3) doesn't exclude vesicle transmission. Demonstrate ferroptosis specificity with Ferrostatin rescue (Fig 2d/e).

The Boyden chamber insert that was used in the experiments shown in Figure 2 has a 3 μm pore size (we have added this information in the legend of Figure 2 in Page 29). Therefore, the results suggested that vesicles with a size below 3 μm in diameter are unlikely involved in the process. We agree with the reviewer that the experiment settings cannot exclude vesicles when their sizes are close to or above 3 μm in diameter. However, neutrophils, which are about 10 μm in diameter, release vesicles that are usually smaller than 3 μm in diameter.

We have performed experiments and demonstrated that the tumor cell death induced by dHL-60 cells can be rescued by ferrostatin-1, liproxstatin-1 and the iron chelator DFO (Revised Figure EV1C). These results confirmed our previous observation that neutrophil- (dHL-60, d32DLC3 and TAN) mediated cell killing can be inhibited by ferrostatin-1, liproxstatin-1 and DFO (Figure

4b and Supplementary Figure 4b in (Yee *et al.*, 2020)). In addition, our previous studies also showed that neutrophil- (dHL-60 and d32DLC3) induced ROS increase in LN229^{TAZ(4SA)} tumor cells can be blocked by ferrostatin-1, liproxstatin-1 and DFO (Supplementary Figures 4d, 4g-h in (Yee *et al.*, 2020)).

Figure 3: HL-60 cells are known to upregulate integrin and adhesion molecules upon differentiation. Using a total integrin blocker might affect multiple adhesion pathways, not just the molecule responsible for this contact. An integrin knockout HL-60 cell line is required for precise elucidation. If non-specific cell contact is crucial, and not a particular integrin, then RGDS-treated neutrophils should still affect tumor killing. However, figures 3g and 3h show only a 20% decrease in tumor cell uptake of neutrophil matter but a dramatic increase in tumor cell viability (fig 3j), suggesting more than just contact is involved.

Among upregulated integrins in HL-60 cells upon differentiation, ITGAV can bind to its ligands through recognizing the RGD motif. Therefore, we have knocked down ITGAV in HL-60 cells through two shRNAs. The ITGAV-depleted dHL-60 cells showed reduced cell killing ability when co-cultured with LN229^{TAZ(4SA)} tumor cells (revised Figure 3L), suggesting that ITGAV is involved in mediating the killing.

In Figure 3g, we showed that the frequency of PKH26+ tumor cells dropped from 82% to 54% upon RGDS treatment, whereas in Figure 3i, we showed the frequency dropped from 59% to 17%. The difference of the frequency drop or fold changes (34% vs 71%) in between these two results could be due to different quantification methods used (manual count in figure 3g vs flow cytometry in figure 3i). Because manual count may not cover all cells in the culture, it may not be as accurate as flow cytometry. Considering this, the effect of RGDS on blocking tumor cell uptake of neutrophil matter would be much higher than 20%.

Figure 4: Why would neutrophils 'commit suicide' in this process if they initiate it to be consumed? Could it involve aged neutrophils or a subtype? There needs to be proof that the dense EM granules are of neutrophil origin and contain MPO.

We have analyzed the differentially expressed genes in between undifferentiated HL-60 and differentiated HL-60 through ingenuity pathway analysis. As shown in revised Figure 3E, top-downregulated pathways are cell cycle-related pathways, whereas top-upregulated pathways are mostly related to terminal differentiation. These results are consistent with previous reports when inducing HL-60 differentiation using similar methods. Therefore, we agree with the reviewer that the property of committing suicide described in this study may associate with terminally differentiated or aged neutrophils.

Our observations indicated that the dense EM granules can be readily found in isolated neutrophils, but not in tumor cells when they are cultured alone. Once tumor cells have engulfed neutrophils, these granules can be seen in tumor cells. These observations suggest that the dense EM granules are of neutrophil origin. To further examine this notion, we have performed immuno-gold TEM to examine MPO. In the coculture containing LN229^{TAZ(4SA)} tumor cells and dHL-60 neutrophils, MPO-immune-gold particles were found in isolated neutrophils, but not in tumor cells when they have no internalized dHL-60 cells. In those LN229^{TAZ(4SA)} cells containing internalized neutrophils, we found MPO-immune-gold particles in the areas corresponding to the fragmented neutrophils (revised Fig. 4G). These MPO-immune-gold particles mostly associate with certain electron-dense granules, which were likely released from the fragmented neutrophil (revised Fig. 4G, insets). These results provided evidence to support that certain dense EM granules are neutrophil origin and contain MPO.

The inhibition of MPO or depletion of Vps34 in an orthotopic xenograft GBM mouse model reduced necrosis formation and increased the survival of tumor-bearing mice. However, these effects may not be mediated by internalized neutrophils and are unrelated to LAP.

We agree with the reviewer that the *in vivo* experiment results may be due to multiple factors, although the results were consistent with our hypothesis. We have added a note of this caveat in the discussion in revised manuscript Page 14, as “It is important to note that *in vivo* tumorigenesis experiments inherently entail various influencing factors, warranting cautious interpretation of the results presented here.”

It's hypothesized that myeloperoxidase (MPO)-containing granules in tumor cells originate from internalized neutrophils. However, no data supports this. The proportion of MPO-containing granules in tumor cells derived from this route is unclear. It is also not clear how MPO-containing granules are produced by the internalized neutrophils.

When observed myeloperoxidase (MPO)-containing granules in tumor cells, we considered several possibilities for the origin. One possibility is free MPO-containing granules or free MPO itself released from neutrophils, which may or may not adhere to tumor cells. An alternative one is internalized neutrophils. As discussed in addressing the above comment regarding Figure 2, our results suggested that vesicles with a size below 3 μm in diameter are not involved in the process and unlikely to be the MPO origin. Therefore, it is unlikely that MPO-containing granules or MPO itself are released from neutrophils and then internalized by tumor cells because their sizes are unlikely larger than 3 μm in diameter. In this case, the only alternative origin other than internalized neutrophils is the neutrophils attached to tumor cells. In this case, the route for MPO-containing granules to transfer from neutrophils to tumor cells could be through establishing a channel in between these cells. However, our EM results have not found an evidence for this. On the contrary, we frequently observed that whole neutrophils are

internalized into tumor cells through regular light microscope (revised Figure 4A-C, 5D, 6A and 6M) and EM (revised Figure 4E, EV3B and EV3D). The internalized neutrophils contain MPO, which can be observed by immunofluorescent staining (revised Figure 1J) and immune-gold TEM (revised Figure 4G). These results provided evidence to support that the major origin is from internalized neutrophils, although we cannot rule out other routes such as neutrophils attached to tumor cells.

When the cell internalization process was blocked by statins, GGTI-298, or Vps34-depletion/inhibition, we found that MPO-containing granules in tumor cells were much fewer (revised Figure 1J, 1K, 6H, 6J and 6L). These results indicated that majority of MPO-containing granules in tumor cells derived from internalized neutrophils.

MPO is produced during the promyelocytic stage, but not in terminally differentiated neutrophils (Nauseef *et al.*, 1988). Synthesized MPO is stored in azurophilic granules during neutrophil differentiation. Since internalized neutrophils are likely to be terminally differentiated, based on our gene expression analysis as discussed in addressing the comment above for Figure 4, internalized neutrophils may not produce MPO but carry the pre-produced MPO from the non-differentiated stages.

The study presents no data to show that HL-60 cell-induced tumor cell death is mediated by MPO-containing granules.

We have examined the transfer of MPO-containing granules in response to simvastatin, fluvastatin, GGTI-298 and YM-53601. The results showed that simvastatin, fluvastatin and GGTI-298, but not YM-53601, can block the transfer of MPO-containing granules (revised Figures 1J and 1K). These effects correlated with their effects on the cell killing ability by dHL-60 (revised Figures 1C, 1E, and EV1A), suggesting an association between MPO-containing granules and the cell killing. Our previous studies showed that knockdown of MPO in dHL-60 cells blocks the neutrophil-induced cell killing (Yee *et al.*, 2020). To further examine whether MPO activity is required for neutrophil-mediated cell killing, we have used 4-aminobenzoic acid hydrazide (4-ABAH) and PF06282999, two small-molecule MPO inhibitors. Adding either of them can rescue LN229^{TAZ(4SA)} cells from dHL-60-induced cell killing (revised Figure. EV1D), therefore confirming the requirement of MPO activity in the cell killing process.

HL-60 and 32dlc3-derived "neutrophils" differ significantly from primary neutrophils.

We have isolated primary neutrophils from LN229^{TAZ(4SA)} cells-derived tumors and used these primary cells to examine and confirm several key conclusions. The results using these cells have been included in revised Figures 1F, 1G, 3K, 6E, 6G, 6H-L, and 6P.

Referee #3:

This work represents a significant addition to the emerging studies in the field, which suggest that tumor-associated neutrophils contribute to glioblastoma progression by increasing intra-tumoral necrosis.

In the present study, the authors performed an unbiased small-molecule screen and found that statins reduce a pro-necrotic interaction between neutrophils and LN229 glioblastoma cells. The most effective fluvastatin and atorvastatin strongly inhibited the neutrophil-mediated tumor cell killing. This effect involved activity of the statin-sensitive mevalonate pathway, required the LC3-dependent phagocytosis of neutrophils by tumor cells, and was possibly dependent on ferroptosis (see comments below). Inhibitor of neutrophil myeloperoxidase and RNAi-driven depletion of Vps34-UVRAG signaling molecules in cancer cells both alleviated the neutrophil-driven glioblastoma necrosis. The same treatments extended animal survival in an orthotopic xenograft tumor model.

Overall, the manuscript makes a strong impression. This study adds to our understanding of glioblastoma biology and carries significant therapeutic relevance. It is expected that this work will be of significant interest to the field and the general readership of the EMBO Journal. Nevertheless, there are a few questions that require additional attention.

Major and moderate comments:

[1] Major issue: Perhaps I am missing something, but all critical in vitro and in vivo findings of this work were collected using LN229 glioblastoma cell line. It is not clear in which experiments (if any) the Authors used the patient derived primary GBM tumor cells which are described in Materials and Methods.

The patient derived primary GBM tumor cells were used in the experiment for Figure 4B. We have added the description in Page 8 and Figure 4B legend (Page 31) of the revised manuscript.

[2] Moderate concern: The Authors propose ferroptosis as the main mechanism for the neutrophil-induced glioblastoma cell death. I appreciate the previous publication from this group (Yee et al., 2020), which tested the mechanistic contributions of ferroptosis. Yet, in the present study, the evidence for the involvement of ferroptosis is rather rudimentary. I see only the effect of liproxstatin-1 and then it is not entirely conclusive. For this reason, I would advise against mentioning ferroptosis in abstract and the first paragraph of discussion (stay with necrosis). I also suggest re-emphasizing the previous work and the role of ferroptosis as discussion progresses. Perhaps an additional reference to the prior work is also warranted for schematic diagram presented in Fig. 7i.

We have replaced ferroptosis with cell death in abstract and the first paragraph of discussion when describing the current study. We cited our previous publication to mention ferroptosis when introducing the background in abstract and discussing the implication in the diagram Fig. 7I.

Technical comment:

[3] There is at least one instance of referring to wrong figure. On p. 7, while describing EM morphology of internalized granules, the authors refer to Fig. 5g, while the information-in-question is presented in Fig. 4g. Please double-check your text for additional typos of this nature.

We thank the reviewer for catching these mistakes and have corrected them.

Reference:

Nauseef WM, Olsson I, Arnljots K (1988) Biosynthesis and processing of myeloperoxidase--a marker for myeloid cell differentiation. *Eur J Haematol* 40: 97-110

Yee PP, Wang J, Chih SY, Aregawi DG, Glantz MJ, Zacharia BE, Thamburaj K, Li W (2022) Temporal radiographic and histological study of necrosis development in a mouse glioblastoma model. *Front Oncol* 12: 993649

Yee PP, Wei Y, Kim SY, Lu T, Chih SY, Lawson C, Tang M, Liu Z, Anderson B, Thamburaj K *et al* (2020) Neutrophil-induced ferroptosis promotes tumor necrosis in glioblastoma progression. *Nat Commun* 11: 5424

Dear Dr Wei Li,

Thank you for submitting your revised manuscript (EMBOJ-2023-115956R) to The EMBO Journal. Your amended study was sent back to the three referees for their scientific re-evaluation, and we have received detailed comments from all of them, which I enclose below. As you will see, the experts state that the work has been substantially improved by the revisions and they are now broadly in favour of publication.

Thus, we are pleased to inform you that your manuscript has been accepted in principle for publication in The EMBO Journal.

We now need you to take care of a number of issues related to formatting and data presentation as detailed below, which should be addressed at re-submission.

Please contact me at any time if you have additional questions related to below points.

As you might have seen on our web page, every paper at the EMBO Journal now includes a 'Synopsis', displayed on the html and freely accessible to all readers. The synopsis includes a 'model' figure as well as 2-5 one-short-sentence bullet points that summarize the article. I would appreciate if you could provide this figure and the bullet points.

Thank you for giving us the chance to consider your manuscript for The EMBO Journal. I look forward to your final revision.

Again, please contact me at any time if you need any help or have further questions.

Best regards,

Daniel Klimmeck

>> Please add up to five keywords for your study.

>> Adjust the title of the 'Conflict of Interest Statement' to 'Disclosure and Competing Interests Statement'.

>> Author Contributions: Please remove the author contributions information from the manuscript text. Note that CRediT has replaced the traditional author contributions section as of now because it offers a systematic machine-readable author contributions format that allows for more effective research assessment. and use the free text boxes beneath each contributing author's name to add specific details on the author's contribution.

More information is available in our guide to authors.

>> Callouts: add a callout on the main text for Figure panel 5F.

>> Data Availability Section: please deposit the RNAseq dataset in a public data repository and ensure privacy is released. Please update the Author Checklist accordingly.

>> Funding: entry in our online system for Penn State College of Medicine Medical Scientist Training Program (5T32GM118294 to P.Y. through PSU) may need to be corrected, and the Four Diamonds (to PSU) is missing.

>> Plagiarism: please avoid textual redundancy with your 2020 NatComms study (PMID: 33110073).

>> Source Data: my colleague Hannah Sonntag will contact you in a separate message shortly requesting source data deposition for the study.

>> Consider additional changes and comments from our production team as indicated below:

- Figure legends:

1. Please note that both figure and figure legend for figure 5f is missing in the manuscript.
2. Please indicate the statistical test used for data analysis in the legends of figure 3e; EV 2a.
3. Please note that information related to n is missing in the legends of figures 2b-c; EV 1b-d; EV 2c; EV 5h.
4. Although 'n' is provided, please describe the nature of entity for 'n' in the legends of figures 1c, e-i; 2e-h; 3b, d, g, i-l; 5b, h; 6c-g, n-p; EV 1a; EV 2f-g; EV 4b; EV 5e.
5. Please note that the error bars are not defined in the legends of figures 1c, e-i, k-l; 2b-c, e-h; 3b, d, g, i-l; 5b, e, h; 6c-g, i-l, n-p; 7b, e, g, EV 1a-d; EV 2c, f-g; EV 4b; EV 5e, h.
6. Please note that the white arrowheads are not defined in the legend of figure 4c. This needs to be rectified.
7. Please note that the red arrows/arrowheads are not defined in the legend of figure 4d-f, EV 3a-d. This needs to be rectified.

Referee #1:

The authors have gone a long way to address my previous concerns. The new data with primary TANs lends additional support to the authors' concept that TANs can indeed induce necrosis in glioblastoma (GBM) cells following their LAP-mediated engulfment and transfer of granule content, predominantly myeloperoxidase. The revision has further strengthened this interesting manuscript. I have no further comments.

Referee #2:

The revised manuscript has been significantly improved by the additional data. The authors have made a forthright effort to address the criticisms raised in the previous review. I am happy to recommend this for Blood without further revision.

Referee #3:

This work represents a significant addition to the emerging studies in the field, which suggest that tumor-associated neutrophils contribute to glioblastoma progression by increasing intratumoral necrosis.

The authors performed an unbiased small-molecule screen and found that statins reduce a pro-necrotic interaction between neutrophils and LN229 glioblastoma cells. The most effective fluvastatin and atorvastatin strongly inhibited the neutrophil-mediated tumor cell killing. This effect involved activity of the statin-sensitive mevalonate pathway, required the LC3-dependent phagocytosis of neutrophils by tumor cells.

The Authors have addressed conceptual and technical concerns of this reviewer. This work is sound and expected to be of significant interest to the field and the general readership of the EMBO Journal.

The authors addressed the minor editorial issues.

Dear Dr Li,

Thank you for submitting the revised version of your manuscript. I have now evaluated your amended manuscript and concluded that the remaining minor concerns have been sufficiently addressed.

I am pleased to inform you that your manuscript has been accepted for publication in the EMBO Journal.

On a different note, I would like to alert you that EMBO Press offers a format for a video-synopsis of work published with us, which essentially is a short, author-generated film explaining the core findings in hand drawings, and, as we believe, can be very useful to increase visibility of the work. Please see the following link for representative examples and their integration into the article web page:

<https://www.embopress.org/doi/full/10.15252/embo.2019103932>

Best regards,

Daniel Klimmeck

Daniel Klimmeck, PhD
Senior Editor
The EMBO Journal
EMBO
Postfach 1022-40
Meyerohofstrasse 1
D-69117 Heidelberg
contact@embojournal.org
Submit at: <http://emboj.msubmit.net>

>>> Please note that it is The EMBO Journal policy for the transcript of the editorial process (containing referee reports and your response letter) to be published as an online supplement to each paper. If you do NOT want this, you will need to inform the Editorial Office via email immediately. More information is available here: <https://www.embopress.org/transparent->

process#Review_Process